# FASTER REINFORCEMENT LEARNING WITH VALUE TARGET LOWER BOUNDING

## ABSTRACT

We show that an arbitrary lower bound of the maximum achievable value can be used to improve the Bellman value target during value learning. In the tabular case, value learning using the lower bounded Bellman operator converges to the same optimal value as using the original Bellman operator, at a potentially faster speed. In practice, discounted episodic return in episodic tasks and n-step bootstrapped return in continuing tasks can serve as lower bounds to improve the value target. We experiment on Atari games, FetchEnv tasks and a challenging physically simulated car push and reach task. We see large gains in sample efficiency as well as converged performance over common baselines such as TD3, SAC and Hindsight Experience Replay (HER) in most tasks, and observe a reliable and competitive performance against the stronger n-step methods such as td-lambda, Retrace and optimality tightening. Prior works have already successfully applied a special case of lower bounding using episodic return, to a limited number of episodic tasks. To the best of our knowledge, we are the first to propose the general method of value target lower bounding with possibly bootstrapped return, to demonstrate its optimality in theory, and effectiveness in a wide range of tasks over many strong baselines.

## 1 INTRODUCTION

The value function is a key concept in dynamic programming approaches to Reinforcement Learning (RL) (Bellman, 1957). Given a starting state, the value function estimates the sum of all future rewards (usually time-discounted). In temporal difference (TD) learning, the value function is adjusted toward its Bellman target which simply adds the reward of the current step with the (discounted) value of the next state (Sutton & Barto, 2018). This forms the basis of many state of the art RL algorithms such as DQN (Mnih et al., 2013), DDPG (Lillicrap et al., 2016), TD3 (Fujimoto et al., 2018), and SAC (Haarnoja et al., 2018).

The value of the next state is typically estimated or derived from the value function itself, which is being actively learned during training, a process called "bootstrapping". The bootstrapped values can be random and far from the optimal value, especially at the initial stage of training, or with sparse reward tasks where rewards can only be achieved through a long sequence of actions. Consequently, the Bellman value targets as well as the learned values are usually far away from the optimal value (the value of the optimal policy).

Naturally, this leads to the following idea: If we can make the value target closer to the optimal value, we may speedup TD learning. For example, we know that the optimal value is just the expected discounted return of the optimal policy, which always upper bounds the expected return of any policy. For episodic RL tasks, we could use the observed discounted return up to episode end from the training trajectories to lower bound the value target. When the empirical return is higher than the Bellman target, lower bounding brings the new value target closer to the optimal value.

For continuing or non-episodic tasks, it is less clear how a lower bound may be estimated. When a continuing task can return negative rewards, a lower bound may not even exist. One could use the n-step bootstrapped return as a lower bound, but bootstrapped return can overestimate and be greater than the optimal value. It is unclear whether the resulting algorithm will still converge.

---

**Algorithm 1** Value iteration with value target lower bounding

---

**Input:** Finite MDP $p(s', r|s, a)$, convergence threshold $\theta$, a lower bound $f(s)$ of the maximum achievable value $\bar{G}^v(s)$
**Output:** State value $v(s)$
$v(s) \leftarrow 0$
**repeat**
   $\Delta \leftarrow 0$
   $v_p(s) \leftarrow v(s)$
   **for** each state $s$ **do**
      $\hat{v}(s) \leftarrow \max_a \sum_{s', r} p(s', r|s, a)[r + \gamma v_p(s')]$
      $\hat{v}_f(s) \leftarrow \max(f(s), \hat{v}(s))$
      $v(s) \leftarrow \hat{v}_f(s)$
      $\Delta \leftarrow \max(\Delta, |v(s) - v_p(s)|)$
   **end for**
**until** $\Delta < \theta$

---

This work presents a general framework proving that value target lower bounding converges to the optimal value for both the episodic and non-episodic cases, under certain conditions for the lower bound function. We demonstrate faster training with an illustrative example and extensive experiments on a variety of environments over strong baselines.

## 2 THEORETICAL RESULTS FOR THE TABULAR CASE

Here we show for the tabular case, arbitrary functions below a certain bootstrap bound can be used to lower bound the value target to still converge to the same optimal value.

### 2.1 BACKGROUND

In finite MDPs with a limited number of states and actions, a table can keep track of the value of each state. Using dynamic programming algorithms such as value iteration, values are guaranteed to converge to the optimum through Bellman updates (Chapter 4.4 (Sutton & Barto, 2018)).

The core of the value iteration algorithm (Algorithm 1) is the Bellman update of the value function, $\mathcal{B}(v)$, where $v(s')$ is the bootstrapped value:

$$\mathcal{B}(v)(s) := \max_a \sum_{s', r} p(s', r|s, a)[r + \gamma v(s')] \tag{1}$$

Here $a$ is an available action in state $s$. $s'$ is the resulting state, and $r$ the resulting reward, of executing $a$ in $s$, with $p(s', r|s, a)$ being the transition probability.

It is well known that the Bellman operator, $\mathcal{B}$, is a contraction mapping over value functions (Denardo, 1967). That is, for any two value functions $v_1$ and $v_2$, $||\mathcal{B}(v_1) - \mathcal{B}(v_2)||_\infty \leq \gamma ||v_1 - v_2||_\infty$ for the discount factor $\gamma \in [0, 1)$ and $||x||_\infty := \max_i |x_i|$ (the $L_\infty$ norm). This guarantees that any value function under the algorithm converges to the optimal value $\mathcal{B}^\infty(v) = v^*$.[1]

### 2.2 CONVERGENCE OF VALUE TARGET LOWER BOUNDING

**Definition 2.1.** The expected n-step bootstrapped return for a given policy $\pi$ and value function $v(s)$ is defined as the expected bootstrapped return of taking $n$ steps according to policy $\pi$:

$$G_n^{\pi, v}(s_0) := \mathbb{E}^\pi \{r_1 + ... + \gamma^{n-1} r_n + \gamma^n v(s_n)\} \tag{2}$$

Here, the step rewards $r_i$ and the resulting n-th step state $s_n$ are random variables, with the expectation $\mathbb{E}^\pi$ taken over all possible n-step trajectories under the policy $\pi$ and the given MDP.

---

[1] For the gist of the proof, see for example page 8 of `https://people.eecs.berkeley.edu/~pabbeel/cs287-fa09/lecture-notes/lecture5-2pp.pdf`

**Definition 2.2.** Given the current learned value function $v(s)$, policy class $\Pi$, the *maximum achievable value* of a state $s$ is defined as:

$$\bar{G}^v(s) \coloneqq \max_{\pi \in \Pi, n \in [1, +\infty)} G_n^{\pi, v}(s) \tag{3}$$

This is a tight definition of maximum because for each state $s$, a different policy $\pi(s)$ and a different number of steps $n(s)$ can be used to achieve the maximum $\bar{G}^v(s)$. It is the maximum achievable bootstrapped return of any single policy or any mixture of policies for any number of steps.

The theorem below says any function not exceeding the maximum achievable value can be used to lower bound the value target, and still achieve the optimal value in convergence.

**Theorem 2.3.** *Under the same assumptions for Bellman value contraction, for any function $f$ that lower bounds the maximum achievable value, i.e. $\forall s, f(s) \leq \bar{G}^v(s)$, if we define the lower bounded Bellman operator as $\mathcal{B}_f(v) \coloneqq \max(\mathcal{B}(v), f)$, then $\mathcal{B}_f^\infty(v) = \mathcal{B}^\infty(v)$.*

Note, the value $v(s)$ and the bootstrapped value can be inaccurate, and even above the optimal value. As a consequence, when $n$ is finite, the maximum achievable value $\bar{G}^v(s)$ (and $f$) can be above the maximum expected return (i.e. the optimal value). On the other hand, when $n$ is sufficiently large, the effect of the bootstrap value $v(s_n)$ diminishes (see Equation 2), and the maximum achievable value becomes the maximum expected return (i.e. the optimal value). Therefore, $\forall s, \bar{G}^v(s)$ is no smaller than the optimal value $\mathcal{B}^\infty(v)(s)$.

As a special case of the theorem, as long as $f$ is below the optimal value, value target lower bounding converges correctly:

**Corollary 2.4.** *If function $f$ lower bounds the optimal value, i.e. $\forall s, f(s) \leq \mathcal{B}^\infty(v)(s)$, then $\mathcal{B}_f^\infty(v) = \mathcal{B}^\infty(v)$.*

A few things to note about the proof of Theorem 2.3 (included in Appendix A.1).

First, this only proves convergence, not contraction under the original $||v_1 - v_2||_\infty$ metric. In the case of the Bellman operator, contraction shows that $\forall v_1, v_2$ value functions, $||\mathcal{B}(v_1) - \mathcal{B}(v_2)||_\infty \leq \gamma ||v_1 - v_2||_\infty$. Here, for value target lower bounding, what's proved is convergence to $v^*$ at a rate of $\gamma$, not contraction. There can be counter examples where the distance between $v_1$ and $v_2$ under one application of $\mathcal{B}_f$ can increase in the original $L_\infty$ metric space, even though $v_1$ and $v_2$ are both getting closer to $v^*$ at a rate of $\gamma$. One difficulty caused by convergence instead of contraction is that the stopping criterion in Algorithm 1 ($\Delta < \theta$) no longer works, due to the inaccessible $v^*$ during learning. In practice, this may not be a serious concern, as people often train algorithms for a fixed number of iterations or time steps.

Second, based on the proof, the new algorithm is at least as fast as the original. When the lower bound actually improves the value target, i.e. $f(s) > \mathcal{B}(v_1)(s)$, there is a chance for the convergence to be faster. Convergence is strictly faster when the lower bound $f$ has an impact on the $L_\infty$ distance between the current value and the optimal value, i.e. it increases the value target for the states where the differences between the current value and the optimal value are the largest.

Third, the lower bound function doesn't have to be static during training. As long as there is a single $f$ during each training update, convergence is preserved.

The following sections detail how to compute lower bounds of the maximum achievable value (Section 3), how to integrate the lower bounds into state of the art RL algorithms (Section 4), and provide an illustration of how this method may benefit value learning in practice (Section 4.3).

## 3 EXAMPLE LOWER BOUND FUNCTIONS

We show a few cases where lower bound functions can be readily obtained from the training experience. Future work may investigate alternatives.

### 3.1 Episodic tasks

In episodic tasks, discounted return is accumulated up to the last step of an episode. During training, we can wait until an episode ends, and compute discounted returns for all time steps. To make it more efficient, we compute and store the discounted return for each time step into the replay buffer, and read it out when the experience is used in training, which adds very little computation to the baseline one-step TD learning.

$$f(s_0) = \sum_{i=0,..,\infty} \gamma^i r(s_i, a_i) \tag{4}$$

We call this variant "lb-DR", short for lower bounding with discounted return.

#### 3.1.1 Episodic with hindsight relabeled goals

In goal conditioned tasks, one helpful technique is hindsight goal relabeling (Andrychowicz et al., 2017). It takes a future state that is $d$ time steps away from the current state as the hindsight / relabeled goal for the current state. When the goal is reached, a reward of 0 is given, otherwise a -1 reward is given for each time step.

In this case, we know it took $d$ steps to reach the hindsight goal, so the discounted future return is:

$$f(s_0) = \sum_{i=0,..,d-1} -1\gamma^i \tag{5}$$
$$= -1(1 - \gamma^d)/(1 - \gamma)$$

This calculation can be done on the fly as hindsight relabeling happens, requiring no extra space and very little computation.

We call this variant "lb-GD", short for lower bounding with goal distance based return.

Additionally, we can also apply lb-DR and lb-GD together, with discounted episodic return (lb-DR) on the original experience and goal distance based return (lb-GD) on the hindsight experience, giving the "lb-DR+GD" variant, which was used in Fujita et al. (2020).

### 3.2 In general (including non-episodic tasks)

If the task is continuing, without an episode end[2], discounted return needs to be accumulated all the way to infinity. When rewards are always non-negative, one can still use the accumulated discounted reward of the future n-steps to lower bound the value. But accumulated n-step discounted reward is no longer a lower bound when rewards can be negative, in which case, the more general lower bounding with bootstrapped return can be used: given a trajectory of training experience $\tau := < s_0, ..., s_n >$:

$$G_n^v(\tau) := r_1 + \gamma r_2 + ... + \gamma^{n-1} r_n + \gamma^n v(s_n) \tag{6}$$

In this case, two variations are possible: Given a trajectory of length $n$,

1. compute $v(s_i)$ for all $i \in [1, n]$ and take the maximum of all $G_i^v(\tau)$ to obtain a tighter lower bound. We call this variant "lb-b-$n$step":

$$f(s_0) = \max_{i \in [1,n]} G_i^v(\tau) \tag{7}$$

2. only evaluate $v$ on the last (i.e. the $n$th) step and use the $n$th-step bootstrapped return as the lower bound, which involves less compute but results in a looser bound. (When $n$ is large enough, this becomes the lb-DR variant.) We call this variant "lb-b-$n$step-only".

$$f(s_0) = G_n^v(\tau) \tag{8}$$

---

[2] Chapter 3.3 of Sutton & Barto (2018) has more details on episodic vs continuing tasks.

The general bootstrapped returns allow lower bounding to be applied to non-episodic tasks, with a bit extra computation in the case of lb-b-$n$step-only, to evaluate value on the $n$th step. The cost can be high for a large $n$ for the lb-b-$n$step variant, but is still comparable to typical n-step return based methods like td-lambda.

## 4 INTEGRATION INTO RL ALGORITHMS

### 4.1 BACKGROUND

The value target lower bounds can be readily plugged into TD learning based algorithms that regress value to a target, e.g. DQN, DDPG or SAC.

In these algorithms, the action value $q(s, a)$ is learned through a squared loss with the target value $y$. In one step TD return, for a batch $\mathbf{B}$ of experience $\{s, a \rightarrow r, s'\}$, the loss is:

$$\mathcal{L}_q := \sum_{(s,a,r,s') \in \mathbf{B}} |q(s,a) - y|^2 \tag{9}$$

In one step TD return, $y$ is the one step TD return $\hat{q}(s, a, r, s')$:

$$\hat{q}(s, a, r, s') := r(s, a) + \gamma q'(s', \mu'(s')) \tag{10}$$

Here, $q'$ and $\mu'$ are the bootstrap value and policy functions, typically following the value and policy functions in a delayed schedule during training. (They are also called "target value" and "target policy", and are very different from the "value target" $y$ in this paper.)

### 4.2 VALUE TARGET LOWER BOUNDING

With lower bounding, we replace the value target $y$ with the lower bounded target:

$$y \leftarrow \max(f, \hat{q}(s, a, r, s')) = \max(f, r + \gamma q'(s', \mu'(s'))) \tag{11}$$

This way of lower bounding the value target is the same as was done by Fujita et al. (2020), but is subtly and importantly different from lower bounding the $q$ value directly (Oh et al., 2018; Tang, 2020): $q(s, a) \leftarrow \max(f, q(s, a))$, which stays overestimated if $q(s, a)$ initially overestimates.

Fujita et al. (2020) only used episodic return to lower bound the value target, while we introduce the general framework of lower bounding with possibly bootstrapped return.

### 4.3 AN ILLUSTRATIVE EXAMPLE

Figure 1 includes a fairly general example showing how value target lower bounding would improve value learning. Suppose we enhance an off policy algorithm such as DDPG with value target lower bounding (lb-DR), when there is no training experience hitting the target state, no meaningful training happens for the baseline or lb-DR. However, when there is one trajectory hitting the target state, all states along the trajectory will soon be propagated with meaningful return, and nearby states will also enjoy faster learning. As the state space becomes larger and the time horizon longer, a successful trajectory will speed up learning quite a bit.

At the core of the above illustration is this simpler example: Consider a simple chain of $n$ states $s_1$ to $s_n$, with $s_i$ going to $s_{i+1}$. It emits reward 1 and terminates when the agent is at the end of the chain, state $s_n$, and emits reward 0 elsewhere. Suppose we have already collected an experience of traversing the chain once, and start training with zero initial value. The basic value iteration algorithm would back up the reward just one step along the chain for each iteration of training, converging in $n$ iterations. However, value target lower bounding trains much faster. Given the collected experience, we precompute the empirical return of each state $G(s_i) = \gamma^{n-i}$, which is also the optimal value. Thus, one training iteration already populates all states with the optimal value.

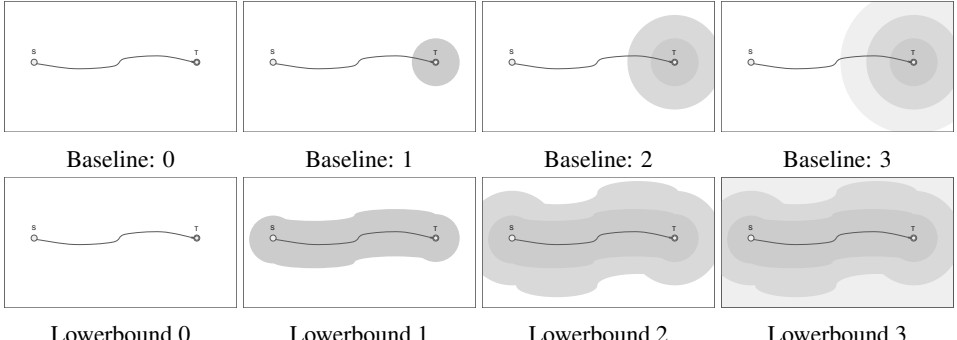

Figure 1: Illustration of value target lower bounding speeding up value learning as training progresses from stages 0 to 3. The task is to navigate in the state space from start state S to end state T, with sparse reward 1 at T and 0 elsewhere. The curve from S to T denotes a training experience that reaches the target. The shaded areas denote states with high value.

## 5 EXPERIMENTS

The goal is to demonstrate the sample efficiency of lower bounding the value target over baseline such as DDPG, TD3, SAC and HER. Because the lower bounded value target can now look many steps into the future, we expect it to be better suited for long horizon, sparse reward tasks. Hence, we choose to experiment on a sampled subset of Atari games, the goal conditioned FetchEnv tasks and the harder goal conditioned Pioneer Push and Reach tasks. See details of the experiment setup in Appendix A.2.

### 5.1 BASELINES

Baselines include DDPG (Lillicrap et al., 2016), TD3 (Fujimoto et al., 2018), SAC (Haarnoja et al., 2018) and HER (Andrychowicz et al., 2017; Plappert et al., 2018). Implementations are based on open sourced repositories, and baseline performance is verified against published results under similar settings. The Appendix A.6 and A.5 include results on more baselines such as DDQN (van Hasselt et al., 2015), td-labmda (Sutton & Barto, 2018) and Retrace (Munos et al., 2016).

### 5.2 HYPERPARAMETERS

Value target lower bounding is applied on top of these baselines without any additional hyperparameter (Section 4). The only hyperparameters come from the baselines. These hyperparameters follow published work as much as possible. When baseline hyperparameters need to be tuned for an environment, e.g. Atari games or Pioneer tasks, we search for the best performance in total episode reward averaged across all tasks for that environment on one set of random seeds, then the optimal hyperparameters are fixed and evaluated on a separate set of random seeds never seen during development. Value target lower bounding simply uses the the parameter values optimal for the baselines. Details are in Appendix A.3.

### 5.3 RESULTS

We report results on both episodic and continuing/non-episodic tasks. We report evaluation performance averaged across several runs of the algorithms (five for the less stable Atari games and three for the others). Each run uses a random seed never seen during development. Due to space constraints, the main paper only reports performance aggregated across all tasks for each environment. During each run, we take one task and one random seed, run baseline and treatment algorithms, and record whether treatment agent evaluates strictly above the baseline agent as training progresses. We average across all the runs of the same environment, and plot the fraction of times where treatment is above baseline and when it is statistically significant in Figure 2. Appendix A.4 contains per task evaluation curves.

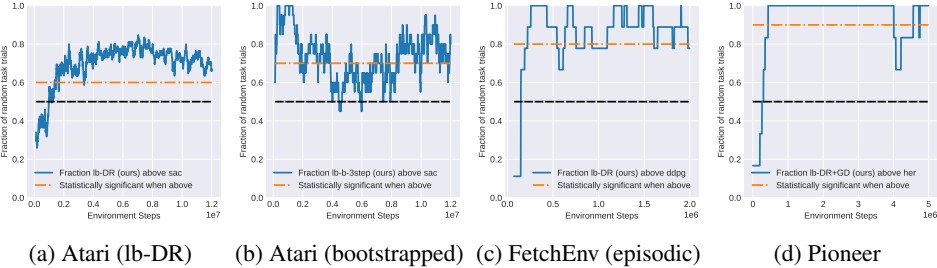

(a) Atari (lb-DR)  (b) Atari (bootstrapped)  (c) FetchEnv (episodic)  (d) Pioneer

Figure 2: Aggregated evaluation performance: The fraction of times where treatment performs strictly above baseline, plotted along the number of time steps used for training. The fraction being above 0.5 means treatment is more often better. We use, for Atari (lb-DR), 85 runs – 17 games each with 5 seeds, for Atari (bootstrapped), 20 runs – 4 games 5 seeds, for episodic FetchEnv, 9 runs – 3 tasks 3 seeds, and for Pioneer, 6 runs – 2 tasks 3 seeds. Statistical significance is calculated with one sided sign test at significance level $p < 0.05$.

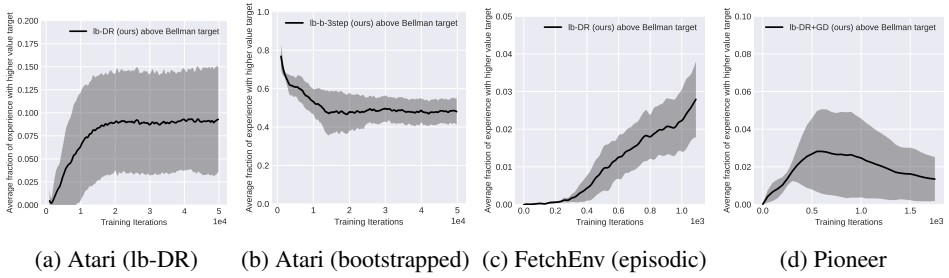

(a) Atari (lb-DR)  (b) Atari (bootstrapped)  (c) FetchEnv (episodic)  (d) Pioneer

Figure 3: Average fraction of training experience where lower bounding improves Bellman value target, plotted along the number of iterations of training. The solid curve is the average of the fraction across all runs – number of tasks times number of seeds, and the shaded area is +/- one standard deviation. Other setups are the same as Figure 2.

### 5.3.1 LOWER BOUNDING VS BASELINES SAC/DDPG/HER

Figure 2 compares the lower bounding treatment with SAC, DDPG or HER baseline on 17 sampled Atari games, the FetchEnv tasks and the Pioneer tasks. For all the environments, value target lower bounding is not only more sample efficient, but also enjoys a higher converged performance. After training starts, it quickly gains ground and outperforms the baseline for 70% to 100% of the runs. It keeps that advantage even at the end of the training, outperforming baseline in converged performance. These plots show most of the times treatment is above baseline, and during most of the training process, the improvement is statistically significant.

Appendix A.4 shows the magnitude of change with total episode return plotted for each task. There is often large gains in sample efficiency and sometimes much higher converged performance. Among all the 22 tasks, only one task (Atari Breakout) shows lb-DR underperforming the baseline.

Investigations show that the loss on Atari Breakout is likely due to the mismatch between training objective and evaluation metric. During training, raw rewards are clipped to [-1, 1] and step discounted at $\gamma = 0.99$ to compute value, while in evaluation, total reward is the unclipped and undiscounted sum of all episode rewards. The discount and clipping together severely penalizes large rewards earned later in the episode, which is what's happening for Breakout, because hitting a top layer block produces a reward of 7 while hitting a bottom layer block produces only 1. When we use non-clipped rewards or a higher $\gamma$ in training, the lower bounding method performs much better in total reward. This train-test discrepancy and another bias of not discounting the state distribution (Thomas, 2014) are likely present in prior works using policy gradient methods on Atari games.

### 5.3.2 VALUE TARGET IMPROVEMENT

The lb-DR method is mostly effective, but is it really due to improvements to the value targets? Figure 3 looks at the fraction of training experience where lower bounding actually improves the Bellman value target over the course of training. Overall, improved value target roughly coincides with performance gains. Appendix A.4 shows the plots per each task.

The Appendix also has more results, comparing with baselines such as n-step methods, DDQN and optimality tightening, and more analyses such as ablations and robustness to hyperparameter choices.

## 6 RELATED WORKS

Prior works (Fujita et al., 2020; Hoppe & Toussaint, 2020; He et al., 2017; Oh et al., 2018; Tang, 2020) employ several different ways of computing future returns and using that as a lower bound to improve value learning. It is quite easy to introduce biases and inefficiencies into the process and end up with a suboptimal or inefficient algorithm, e.g. see discussion below about Self Imitation Learning and episodic control (Oh et al., 2018; Tang, 2020; Blundell et al., 2016), or n-step methods. Our work is the first to propose the general form of value target lower bounding (possibly with bootstrapping) for both the episodic and non-episodic tasks, to show its convergence to the optimal value in the tabular case, and to demonstrate its effectiveness in extensive experiments on a wide range of tasks.

Fujita et al. (2020)'s method is similar to a special case (the lb-DR+GD variant) of the general method. It was used as a part of a large system and was shown to improve sample efficiency for a robotic grasping task. Hoppe & Toussaint (2020) also bound the value target. But instead of using empirical return, they use a simplified MDP with a subset of actions. Though without theoretical proof and only experimented on a limited set of tasks, both works show that value target lower bounding increases sample efficiency empirically. Our work, in addition to the theory and the more general method, shows that lower bounding improves both sample efficiency and converged performance in a wide range of tasks.

Optimality tightening (He et al., 2017) also uses empirical return with bootstrap to improve value learning, but in a different way. It formulates value learning as a constrained optimization problem using the empirical bootstrapped value to provide the lower and upper constraints of the value function. In the experiments, the Lagrangian multiplier is fixed rather than being learnt, which would likely lead to suboptimal solutions. Our lb-b-$n$step method also uses bootstrapped value. We lower bound the value target directly, which is simpler, more efficient, and likely more optimal. Our framework includes even more efficient and effective methods like lb-DR for episodic tasks. Appendix A.5 offers more discussions and results related to the optimality tightening method.

Our work is subtly but importantly different from the prior works on lower bound Q learning or Self Imitation Learning (SIL) (Oh et al., 2018; Tang, 2020). SIL uses empirical return $R$ to lower bound the value function itself (instead of the *value target*). This is achieved by adding an off policy value loss during on-policy (AC or PPO) training ($L_{value}^{sil} = \frac{1}{2}|v(s) - \max(v(s), R)|^2$). When the value function overestimates, the SIL value loss becomes zero, and keeps overestimating. Mixing the SIL loss with the loss from the baseline algorithms probably helped to correct the overestimation, but no theoretical guarantee was given. In evaluation, SIL was often compared to on-policy Actor Critic or PPO baselines, so it was not clear how much of the gain was due to lower bounding and how much due to off-policy value learning. In this work, we bound the Bellman value target (Equation 11), so overestimates are automatically corrected via Bellman updates, and convergence is guaranteed in the tabular case. We also use off-policy algorithms as baselines for a cleaner comparison.

Episodic control (Blundell et al., 2016) and follow-up works (Lin et al., 2018; Sarrico et al., 2019; Hu et al., 2021; Ma et al., 2022) use episodic memory from past experiences to develop value lower bounds. When multiple experiences start from the same state, the maximum episodic return is stored in the episodic memory slot for that state. This provides a potentially tighter lower bound than the episodic return used in this paper (which only uses the episodic return from the one episode where the transition is sampled). During control, the action that maximizes the stored episodic return is picked. For the general non-tabular case, function approximation is used to represent states, which bootstraps values of states never encountered in training before. The original episodic control and

a few of the variations (Blundell et al., 2016; Lin et al., 2018; Sarrico et al., 2019) lower bound the value function itself, not the value target, similar to how Self Imitation Learning (Oh et al., 2018) does. Hence, an initially overestimated episodic value, due to either improper initialization or function approximation, stays overestimated throughout training, because it never goes through the Bellman operator. These methods may not converge to the optimal value even for tabular deterministic environments. Follow-up works (Hu et al., 2021; Ma et al., 2022) use implicit memory based planning, which essentially lower bound the value target with a function based on episodic return, and avoid the earlier overestimation problem (see Appendix A.8). Maximum entropy Mellowmax episodic control (Sarrico et al., 2019) uses a temperature controlled softmax based operator to generate the action probabilities and is similar to the discrete version of SAC. Overall, episodic memory is an interesting tool to come up with tight value target lower bounds, while the value target lower bounding we propose is more general along two directions: the theory works for stochastic environments, and for non-episodic tasks (with experiments in Appendix A.4.3).

Hindsight Experience Replay (HER) (Andrychowicz et al., 2017) is a related prior work to compare to. HER is simple, effective, efficient, and also relies on the unbiased replay of past experiences, as explained in Appendix A.7. HER works best in sparse reward goal conditioned tasks. Unlike our work, HER additionally requires the task to be goal conditioned, and relies on full knowledge of the reward function to work. HER also has one additional hyperparameter to tune – the proportion of hindsight experience. Our method is more general, applicable to non-goal conditioned tasks and to dense reward scenarios, e.g. Atari games. We employ the orthogonal idea of value target lower bounding to improve RL training, and provide additional significant gains even on top of HER, as shown in the experiments on hard continuous control tasks.

N-step return methods such as td-lambda (Sutton & Barto, 2018) and Retrace (Munos et al., 2016) also look a few steps ahead to obtain a better value. Traditionally, this requires careful off-policy correction, and the value can still be far from the optimal value due to the often suboptimal behavior policy. This work shows that value target lower bounding efficiently and effectively looks ahead much further without the need for off-policy correction. Lower bounding with episodic return is also more efficient, without having to compute value of all n-steps. Appendix A.5 has more detailed observations and discussions.

## 7 CONCLUSIONS

Overall, value target lower bounding is a simple method without any additional hyperparameters to tune. It is efficient, highly effective, and theoretically justified. It can be readily applied to any TD learning based RL algorithms, such as DQN, DDPG and SAC. In theory, value target lower bounding converges to the same optimal solution as the original Bellman operator, at a speed at least as fast. In practice, extensive experiments across a wide range of tasks show that it greatly improves sample efficiency, and in several cases even converge to a higher performance.

For episodic tasks, discounted episodic return lower bound is efficient and effective, achieving much higher sample efficiency and converged performance than one-step baselines such as SAC, DDPG or TD3 in most tasks, and is competitive among n-step baselines. For goal conditioned tasks, simple goal distance based return achieves higher value than Hindsight experience relabeling (HER) in difficult long horizon tasks.

For non-episodic tasks or in general, lower bounding with n-step bootstrapped return outperforms one-step baselines and is a strong competitor to the n-step methods such as (truncated) td-lambda and Retrace.

### 7.1 FUTURE WORK

There could be better value lower bounds that improve training even more. One such direction is to use planning (e.g. Monte Carlo Tree Search, the Cross Entropy Method or using subgoals) to achieve tighter lower bounds with the help of a model of the task.

Other ways of bounding the value target may be worth investigating as well. An example is value upper bounding, to reduce value overestimation. However, given the disparate motivation from lower bounding, which is to speed up value improvement, we leave upper bounding to future work.

ETHICS STATEMENT

This research belongs to basic reinforcement learning, accelerating the training of reinforcement learning algorithms. Helpful agents trained on benevolent tasks will learn faster, so will malicious agents trained to do damage. All the usual societal impact of reinforcement learning algorithms apply. The reader and user has to use caution and their own judgments when applying these automation algorithms to the real world: careful testing and scaling is needed, starting from virtual simulations, to testing in a lab environment, to small scale real world tests, and eventually to full scale deployment with careful monitoring in place.

REPRODUCIBILITY STATEMENT

Our code change is based on a publicly available RL library, with strong baselines already implemented. Our relatively small code change is committed to a private github repository, which we plan to open source upon publication. Experiment parameters are configured and controlled by an automation script, with each experiment label corresponding to the set of configurations used for that experiment, so there is little room for manual error when running many experiments across different tasks, methods and hyperparameters. When running experiments, the snapshot of the code used to run each experiment is stored together with the results for verification.

Experiments are done in simulation with pseudo randomness. We've run our code on different machines with different GPU hardware using the same docker image, and the results are reproducible up to every float number using the same random seed. In a few cases, we've also run our code on different versions of hardware and software (CUDA and pytorch), and the results are similar, though not the same at the float number level.

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

# A    APPENDIX

## A.1    PROOFS

Before going on to Theorem 2.3, we first prove the easier Corollary 2.4.

We want to show that given a lower bound of the optimal value, $\forall s, f(s) \leq \mathcal{B}^{\infty}(v)(s)$, under the new operator $\mathcal{B}_f$, the value function converges to the same optimal value function given by the Bellman operator $\mathcal{B}$.

*Proof.* Let $v^*$ be the fixed point and optimal value of the original Bellman operator: $v^* := \mathcal{B}^{\infty}(v)$, $v$ be any value function, and $s$ any state.

First, for the simple case of $\forall s$, where $f(s) \leq \mathcal{B}(v)(s)$, the new operator backs off to the Bellman operator, and follows the convergence of the Bellman operator:

$$
\begin{aligned}
&|\mathcal{B}_f(v)(s) - v^*(s)| \\
=&|\max(\mathcal{B}(v)(s), f(s)) - v^*(s)| \\
=&|\mathcal{B}(v)(s) - v^*(s)| \\
=&|\mathcal{B}(v)(s) - \mathcal{B}(v^*)(s)| \\
\leq&\gamma||v - v^*||_{\infty}
\end{aligned}
$$

Second, $\forall s$, where $f(s) > \mathcal{B}(v)(s)$,

$$
\begin{aligned}
&|\mathcal{B}_f(v)(s) - v^*(s)| \\
=&|\max(\mathcal{B}(v)(s), f(s)) - v^*(s)| \\
=&|f(s) - v^*(s)| \\
=&v^*(s) - f(s) \quad (\text{because } f \text{ lower bounds } v^* : v^*(s) \geq f(s)) \\
<&v^*(s) - \mathcal{B}(v)(s) \quad (\text{because } f(s) > B(v)(s)) \\
=&|\mathcal{B}(v)(s) - v^*(s)| \\
\leq&|\mathcal{B}(v)(s) - v^*(s)| \\
=&|\mathcal{B}(v)(s) - \mathcal{B}(v^*)(s)| \\
\leq&\gamma||v - v^*||_{\infty}
\end{aligned}
$$

Therefore, the distance to the optimal value shrinks by $\gamma$ with every application of $\mathcal{B}_f$:

$||\mathcal{B}_f(v) - v^*||_{\infty} = \max_s |\mathcal{B}_f(v)(s) - v^*(s)| \leq \gamma||v - v^*||_{\infty}.$

According to the definition of convergence to $v^*$, we need to find an $N$, such that $\forall \epsilon > 0, \forall v \neq v^*$, $\forall n > N, ||\mathcal{B}_f^n(v) - v^*||_{\infty} < \epsilon$.

We can easily calculate that any $N \geq \log_{\gamma} \frac{\epsilon}{||v-v^*||_{\infty}}$ (note, $\gamma < 1$) satisfies the condition, which concludes the proof that any value function $v$ will converge to $v^*$ under the lower bounded Bellman operator $\mathcal{B}_f$. $\qquad \square$

Note, from the proof above, we can see that $\mathcal{B}_f$ converges faster than $\gamma$ (the speed of Bellman contraction), when the lower bound is strictly above the Bellman value target, i.e. $f(s) > \mathcal{B}(v)(s)$.

Now for Theorem 2.3: given the maximum achievable value $\bar{G}^v(s)$ and given that $f(s) \leq \bar{G}^v(s)$, we want to show convergence to the optimal value.

*Proof.* First, $\forall s$, where $f(s) < \mathcal{B}(v)(s)$, the value target backs off to the original Bellman target, and the distance to the optimal value shrinks at rate $\gamma$.

Second, $\forall s$, where $f(s) < v^*(s)$, it follows from Corollary 2.4 that the distance to the optimal value shrinks at rate $\gamma$.

Last, we only need to prove for any $s$, where $f(s) \geq v^*(s)$ and $f(s) \geq \mathcal{B}(v)(s)$, the distance to the optimal value still shrinks:

$$|\mathcal{B}_f(v)(s) - v^*(s)|$$
$$=|\max(\mathcal{B}(v)(s), f(s)) - v^*(s)|$$
$$=|f(s) - v^*(s)|$$
$$=f(s) - v^*(s)$$
$$\leq \bar{G}^v(s) - v^*(s)$$

According to the definition of $\bar{G}^v$ in Equation 3 of the main text:

$$= \max_{\pi \in \Pi, n \in [1, +\infty)} G_n^{\pi,v}(s) - v^*(s)$$

Now suppose $\pi'$ and $n(s)$ achieves the maximum bootstrapped value $\bar{G}^v(s)$:

$$= G_{n(s)}^{\pi',v}(s) - v^*(s)$$

According to the definition of n-step bootstrapped value $G_n^{\pi,v}(s)$ in Equation 2 of the main text:

$$= \mathbb{E}^{\pi'}\{r_1 + \gamma r_2 + ... + \gamma^{n(s)-1} r_{n(s)} + \gamma^{n(s)} v(s_{n(s)})\} - v^*(s)$$

(The expectation above is over all possible $n(s)$-step trajectories of the given policy $\pi'$ and MDP.)

Suppose $\pi^*$ is the optimal policy, which achieves maximum value $v^*$ for any number of steps $n$ and state $s$:

$$= \mathbb{E}^{\pi'}\{r_1 + \gamma r_2 + ... + \gamma^{n(s)-1} r_{n(s)} + \gamma^{n(s)} v(s_{n(s)})\} -$$
$$\mathbb{E}^{\pi^*}\{r_1 + \gamma r_2 + ... + \gamma^{n(s)-1} r_{n(s)} + \gamma^{n(s)} v^*(s_{n(s)})\}$$
$$\leq \mathbb{E}^{\pi'}\{r_1 + \gamma r_2 + ... + \gamma^{n(s)-1} r_{n(s)} + \gamma^{n(s)} v(s_{n(s)})\} -$$
$$\mathbb{E}^{\pi'}\{r_1 + \gamma r_2 + ... + \gamma^{n(s)-1} r_{n(s)} + \gamma^{n(s)} v^*(s_{n(s)})\}$$

(This is because $\pi^*$ maximizes the expected n-step bootstrapped value. So the expected value of any other policy, e.g. $\pi'$, bootstrapped with $v^*$ cannot be greater.)

$$= \gamma^{n(s)} \mathbb{E}^{\pi'}\{v(s_{n(s)}) - v^*(s_{n(s)})\}$$
$$\leq \gamma^{n(s)} \mathbb{E}^{\pi'}|v(s_{n(s)}) - v^*(s_{n(s)})|$$
$$= \gamma^{n(s)} \sum_{s_{n(s)}} \{p^{\pi'}(s_{n(s)}|s) \times |v(s_{n(s)}) - v^*(s_{n(s)})|\}$$
$$\leq \gamma^{n(s)} \max_s |v(s) - v^*(s)|$$
$$\leq \gamma^{\min_s n(s)} \max_s |v(s) - v^*(s)|$$
$$= \gamma^{\min_s n(s)} ||v - v^*||_\infty$$

This means in the case of overestimated bootstrap values, the new operator promises to shrink at a rate of $\gamma^{\min_s n(s)}$, and overall, the new operator will at least shrink at a rate of $\gamma$. $\qquad \square$

Note, from the proof above, we can see that when the lower bound overestimates, i.e. $f(s) \geq v^*(s)$, $\mathcal{B}_f$ converges at a speed of $\gamma^{\min_s n(s)}$, which could be faster than $\gamma$, the speed of Bellman contraction.

These proofs work for stochastic MDPs, because we treat trajectory rewards and states as random variables conditioned on the MDP and the policy. The proofs work for action values as well, by simply replacing the value function above $v(s)$ with the action value $q(s, a)$, and the value lower bound $f(s)$ with the action value lower bound $f(s, a)$.

$\pi'$ in theory can be different for different state $s$, so that when unrolling from state $s_0$ for a few steps into $s_i$, it still follows $\pi'(s_0)$, instead of $\pi'(s_i)$. However, it's easy to prove (by contradiction) that there exists a single policy $\pi'$ which achieves the maximum achievable value (as long as ties are split deterministically).

## A.2 EXPERIMENT SETUPS

We experiment on three sets of tasks with different input characteristics and control difficulty. Some of the tasks are not goal conditioned, so only lower bounding with empirical discounted return is available. Some of them are goal conditioned, so both empirical discounted return and hindsight relabeling with discounted goal return are available as lower bounds.

### A.2.1 ATARI GAMES

We experiment on the classical Atari games with image input to test using discounted episodic return to lower bound value target. We picked the popular games Breakout, Seaquest, Space Invaders (these three were used for hyperparameter tuning by Mnih et al. (2013)), Atlantis, Frostbite and Q*bert (these three were highlighted in (He et al., 2017)), and sampled another 11 games from the total 56 (simply using one out of every five games in alphabetical order). As with prior work (Mnih et al., 2013; Oh et al., 2018), we evaluate on the standard NoFrameskip-v4 versions of the games with actions repeated for a fixed four frames and each game started with up to 30 random number of noop steps before handing to the agent (to initialize the pseudo random number generator differently across episodes). Each episode of the games is capped at 10,000 time steps, with the last time step having discount 0.99 when the time limit is reached, i.e. resetting the game without ending the episode. For any regular game end, e.g. after losing all lives, the episode ends with a last step discount of 0. Because the games are episodic, both lb-DR and lb-b-$n$step methods can be applied.

### A.2.2 FETCH PUSH, SLIDE AND PICKANDPLACE

The FetchEnv tasks (Plappert et al., 2018) are goal conditioned tasks with a MuJoCo simulated robotic arm moving objects on a table. Robot states and object position serve as input. The agent outputs continuous actions taking the form of relative positions to move to. A PID controller translates the relative position actions into the exact torque applied at each joint. Rewards are sparse and goal-conditioned, with -1 for non-goal states and 0 for goal states.

By default the FetchEnv tasks are non-episodic. They reset every 50 steps, but all steps including the step right before task reset have the same positive discount (Andrychowicz et al., 2017). As explained in Section 3.1, to use episodic discounted return as lower bound, we can make them episodic by adding a gym wrapper around the environment to end an episode after its goal is achieved, and reset the task. When a goal is not reached within 50 steps, we just reset the task without ending the episode, as is done in the original FetchEnv, and such experience is not used in value target lower bounding.[3] This also changes the nature of the tasks, so the agent does not have to stay at the goal state indefinitely, but instead only needs to reach the goal position as fast as possible. This makes the episodic FetchEnv tasks slightly easier to train than the original tasks, because the agent only needs to reach the goal state quickly, instead of having to reach and stay at the goal position indefinitely. (There are ways to avoid changing the desired behavior by e.g. including agent's speed into the goal state or requiring the agent to stay at the goal position for several time steps before ending the episode. This seems orthogonal to the main idea here, and is not included in this work.)

We experiment on both the original/non-episodic FetchEnv tasks (with lb-b-$n$step methods) and the episodic FetchEnv tasks (with lb-DR and lb-GD methods).

Compared with the Atari games, the inputs are simpler, no longer image based, but the control task is continuous, under realistic physical simulation and harder.

### A.2.3 PIONEER PUSH AND REACH TASKS

This is a set of challenging goal reaching and object pushing tasks for the car Pioneer 2dx, physically simulated with Gazebo. The car is 0.4 meter long. Objects and goal positions are randomly initialized between 0.5 meter to 1 meter of each other inside a 10 meter by 10 meter flat space. Inputs are the car and object states and the goal positions, and actions are the forces applied on the two driving wheels.

---

[3]Fujita et al. (2020) chose to end episodes when either a maximum of T time steps is reached or the goal is reached, and provided the agent with the number of timesteps since episode start as input to the agent, so that the agent is aware of the approaching episode end.

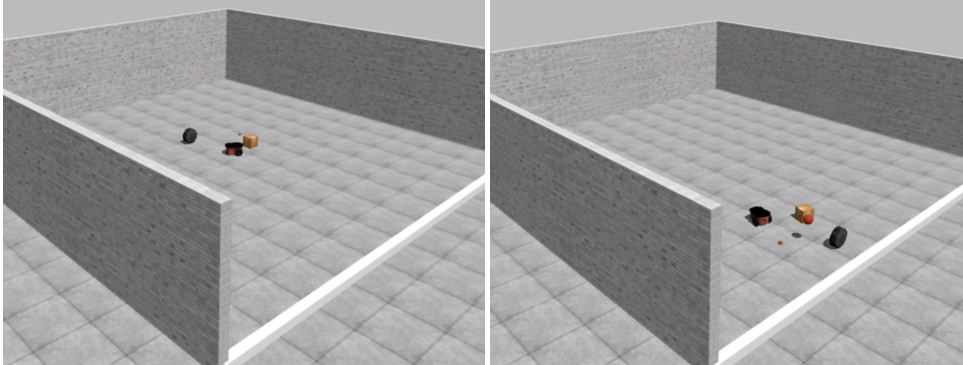

Figure 4: The Pioneer Push task and the Push and Reach task.

For the Pioneer Push task, the car has to push a block to within 0.5 meter of the 2 dimensional goal position indicated by a small red dot on the ground. For the Pioneer Push and Reach task, the car has to first push the object to the goal location (red dot) and then drive to a separate goal position (red ball in the air); the goal is achieved when the concatenation of the two goal locations (for Push and for Reach) is within 0.5 of the concatenated achieved positions (of the block and the car) in $L_2$-distance.

These tasks are episodic with sparse goal reward, and we only test the lb-GD and lb-DR+GD methods on them with HER as baseline. (TD3 without HER takes too long to train.)

These tasks take longer time to accomplish, and also take longer time to train than the FetchEnv tasks. Some of the reasons are the force based wheel control instead of the higher level position control, and the arena space being much larger than just a tabletop.

### A.3 HYPERPARAMETERS AND BASELINES

Table 1 lists the hyperparameters of the baseline algorithms. For FetchEnv, they follow published work (Plappert et al., 2018). For Atari and Pioneer tasks, they are tuned using one set of random seeds and after keeping the hyperparameters fixed, trained with a different set of random seeds and evaluated. Value target lower bounding has no parameter, and we did not re-tune any parameters of the baseline RL algorithms for value target lower bounding. When comparing lb-b-$n$step methods with other n-step methods, we simply use the same $n$ as the other baselines.

For the Atari games, the original DQN setup with only one training environment takes too long to train so we decided to tune SAC as baseline and found it to outperform published Actor-Critic results (Oh et al., 2018) and our tuned DDQN (results in Appendix A.6).

To use SAC on discrete actions, we simply swap out SAC's value networks (which take in continuous action and state as input and produce the value) with DQN's Q networks (which take in state as input and output $n$ heads, each head for one action logit). We pass the output of the Q networks (administered by the temperature $\alpha$) through softmax to compute the action probabilities, and to further sample actions. We ignore actor losses because there is no separate actor network, and only use critic losses to train the Q networks.

The reason we can use Q with temperature and softmax to compute action probabilities is that for discrete actions, maximizing the SAC objective $\max_p \left[ p(a_i, s)(Q(s, a_i) - \alpha \log p(a_i, s)) \right]$ subject to $\sum_i p(a_i, s) = 1$ directly gives the solution $p(a_i, s) \propto \exp(Q(s, a_i)/\alpha)$. See Appendix D of (Yu et al., 2021) for a similar derivation.

For Pioneer Push and PushReach tasks, TD3 is used, (we simply equip DDPG with two critics for clipped double Q learning (Fujimoto et al., 2018)), which works better than DDPG with one critic.

Table 1: Hyperparameters for all the tasks

| Hyperparameters\Tasks | Atari (SAC) | FetchEnv (DDPG) | Pioneer (TD3) |
|---|---|---|---|
| Parallel environments | 30 | 38 | 30 |
| Unrolls per train iteration | 8 | 50 | 100 |
| Updates per train iteration | 4 | 40 | 40 |
| Mini-batch size | 500 | 5,000 | 5,000 |
| Training updates per target network update | 20 | 40 | 40 |
| Target update weight | 0.95 | 0.95 | 0.95 |
| Discount per time step | 0.99 | 0.98 | 0.99 |
| Initial collect steps | 100,000 | 10,000 | 10,000 |
| Total training time steps | 12 million (x4 environment frames) | 2 million | Push: 5 million, PushReach: 14 mil |
| Max steps before task reset | 10,000 | 50 | Push: 100, PushReach: 200 |
| Replay buffer size | 1 million | 2 million | 6 million |
| Adam optimizer learn rate[a] | $5e^{-4}$ | $1e^{-3}$ | $1e^{-3}$ |
| Network structure | conv((32, 8, 4), (64, 4, 2), (64, 3, 1)) + fc(512)[b] | fc(256, 256, 256) | fc(256, 256, 256) |
| Number of critics | 2 | 1 | 2 |
| $\epsilon$-greedy for evaluation | 0.05 | 0.3 | 0.3 |
| Evaluation interval in train iters | 1000 | 40 | 200 |
| Evaluation episodes | 100 | 200 | 100 |
| Life loss as terminal[c] | Yes | n/a | n/a |
| Action repeat | 4 | n/a | n/a |
| Image scaling | [-1, 1] | n/a | n/a |
| Frame stacking | 4 | n/a | n/a |
| Reward clipping | [-1, 1] | n/a | n/a |
| SAC target entropy | calculated[d] | n/a | n/a |
| Hindsight percentage | n/a | 80% | 50% |
| Observation normalization[e] | No | Yes | No |
| TD-lambda: $\lambda$ | 0.95 | 0.95 | n/a |
| $n$-step bootstrap: $n$ | 3 | 2 | n/a |

[a] Adam $\hat{\epsilon} = 1e^{-7}$ for all tasks.

[b] Netowrk structure for Atari follows DDQN (van Hasselt et al., 2015).

[c] Life loss in Atari games is treated as a terminal state in training, following EfficientZero (Ye et al., 2021).

[d] The SAC target entropy is set to the entropy of uniformly distributing 0.1 probability mass across all but one actions.

[e] Observations are normalized to have zero mean and unit variance based on the statistics of the training observations, and we found the normalization to be critical in reproducing HER results on FetchEnv.

### A.4    RESULTS

Applying lower bounding (e.g. lb-DR) on different baseline algorithms e.g. DDPG or SAC results in different treatment methods. Since we always compare treatment with its corresponding baseline, throughout the paper, we simply call the treatment lb-DR etc. without mentioning the baseline algorithm.

#### A.4.1    LB-DR (EPISODIC RETURN) VS BASELINE SAC/DDPG

Figure 5 compares lower bounding with discounted return (lb-DR) against SAC or DDPG baseline on 17 sampled Atari games and the episodic FetchEnv tasks.

For 16 out of the 17 Atari games, lower bounding with episodic discounted return (lb-DR) performs at least as well as the baseline, often much better. On more than half of the Atari games and on the Fetch PickAndPlace task, there are large gains in both sample efficiency and final performance. On FetchPush and a few of the Atari games (Alien, Bank Heist and Fishing Derby), there is about 70% sample efficiency gain with similar converged performance. Among all the 20 tasks, only 1 task (Atari Breakout) shows lb-DR underperforming the baseline.

#### A.4.1.1    Value learning plots

This section presents plots of learned value and how often value is improved by the proposed methods, in order to show the effect of lower bounding on value improvement.

Figure 6 shows the fraction of training experience where lb-DR value target is greater than the Bellman target from SAC/DDPG on the 17 Atari games and the episodic FetchEnv tasks (without hindsight). They correlate well with actual performance (Figure 5) and with how value is learning (Figure 7).

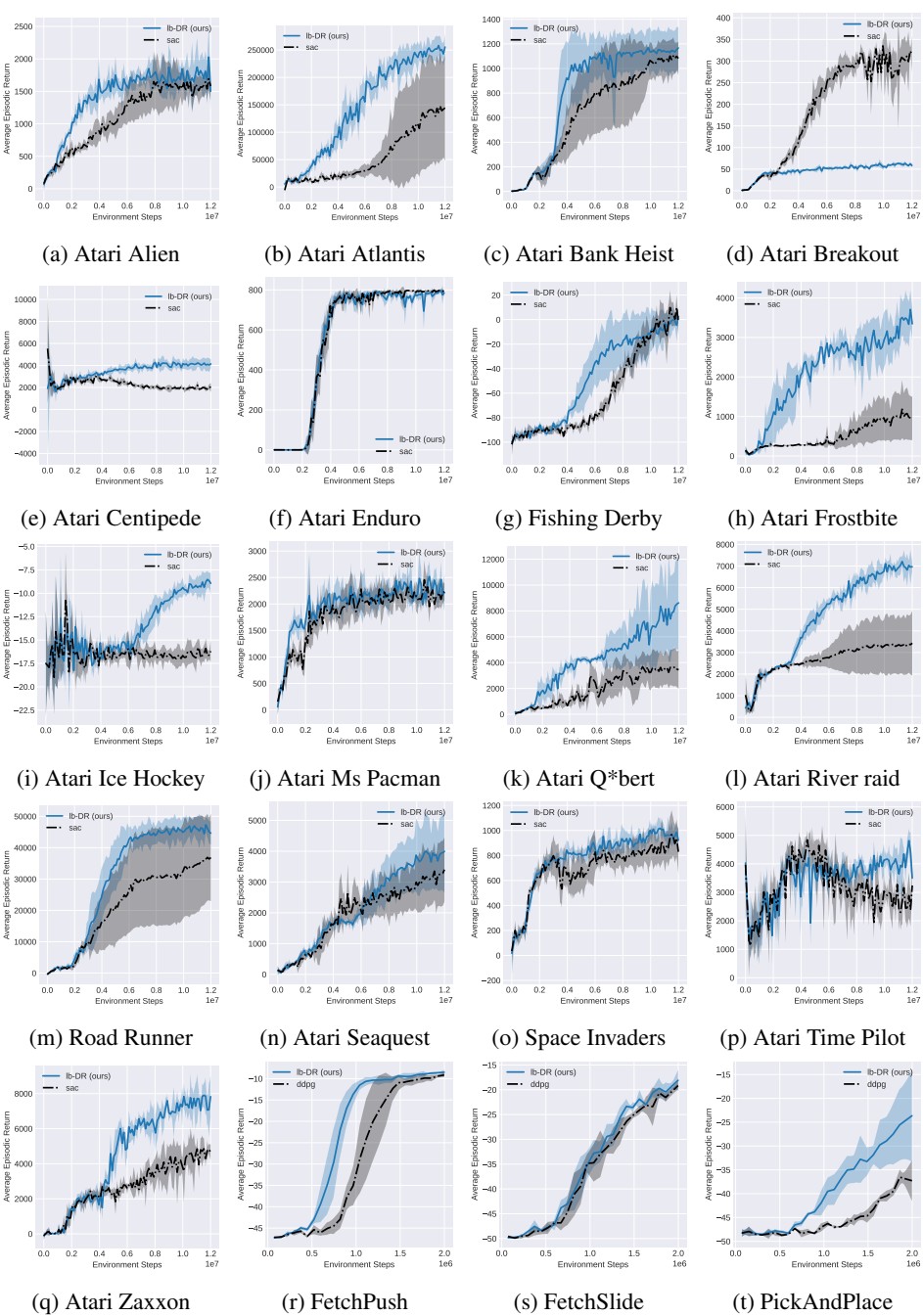

Figure 5: Evaluated average return of value target lower bounding with discounted return (lb-DR) vs SAC or DDPG on Atari games and episodic FetchEnv tasks. Solid curves are the mean across five (for Atari) or three (others) seeds, and shaded areas are +/- one standard deviation.

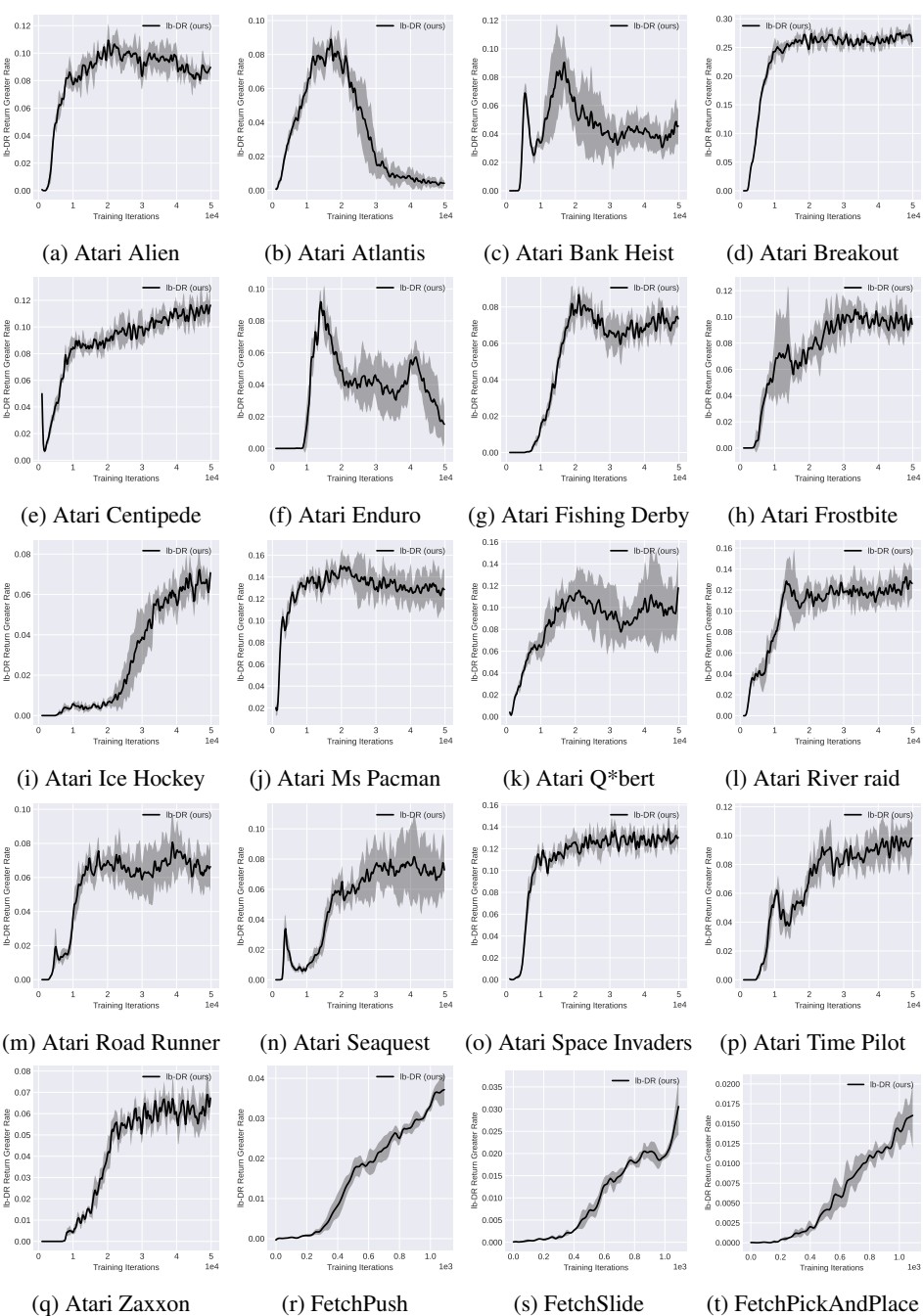

Figure 6: Fraction of training experience where lb-DR value target is greater than the Bellman target, on Atari games and episodic FetchEnv tasks, plotted against the number of training iterations. Solid curves are the mean across five (for Atari) or three (others) seeds, and shaded areas are +/- one standard deviation.

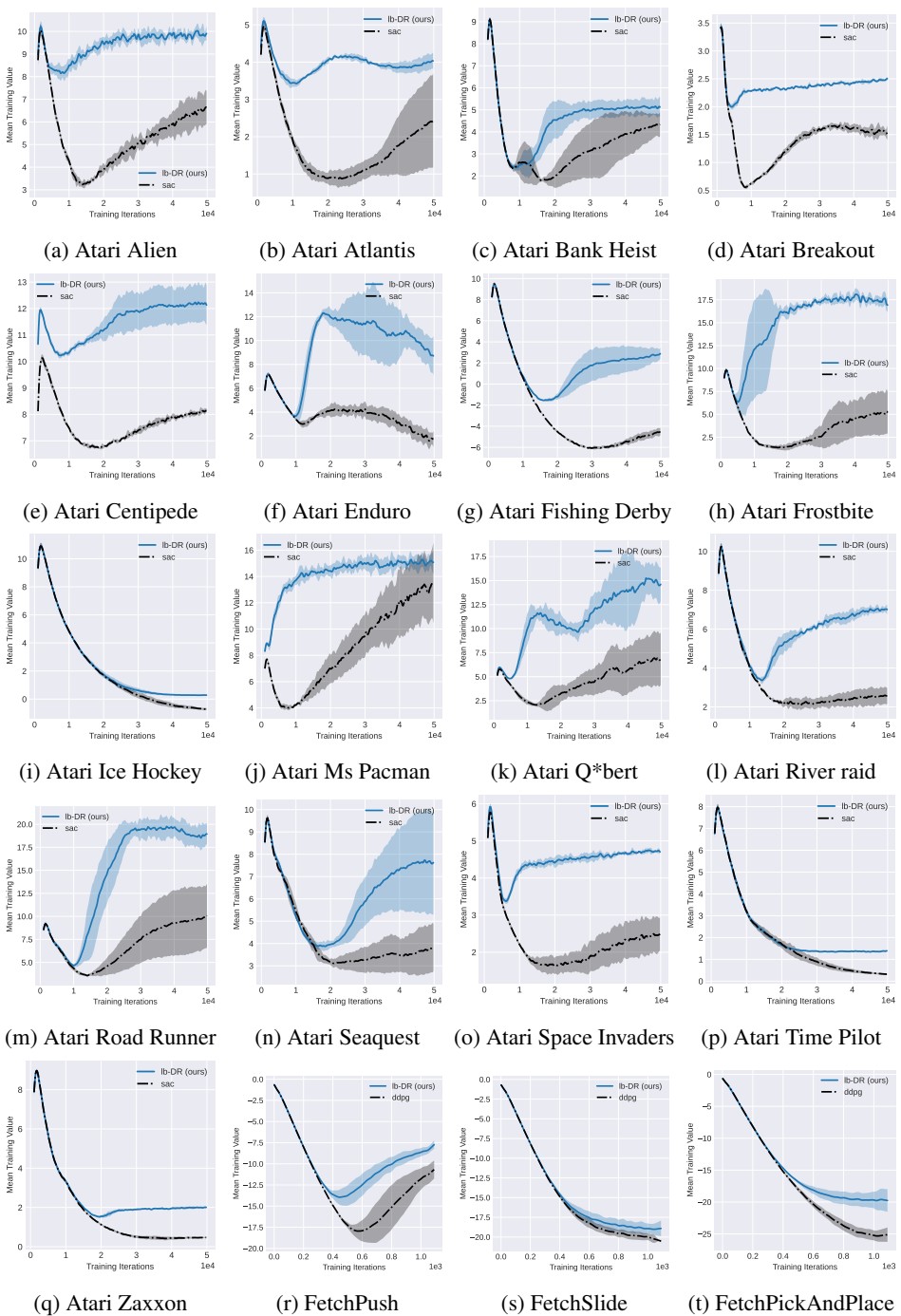

Figure 7: Learned values of lb-DR and SAC (for Atari games) and DDPG (for FetchEnv tasks), evaluated on the training experience and plotted against the number of training iterations. Solid curves are the mean across five (for Atari) or three (others) seeds, and shaded areas are +/- one standard deviation.

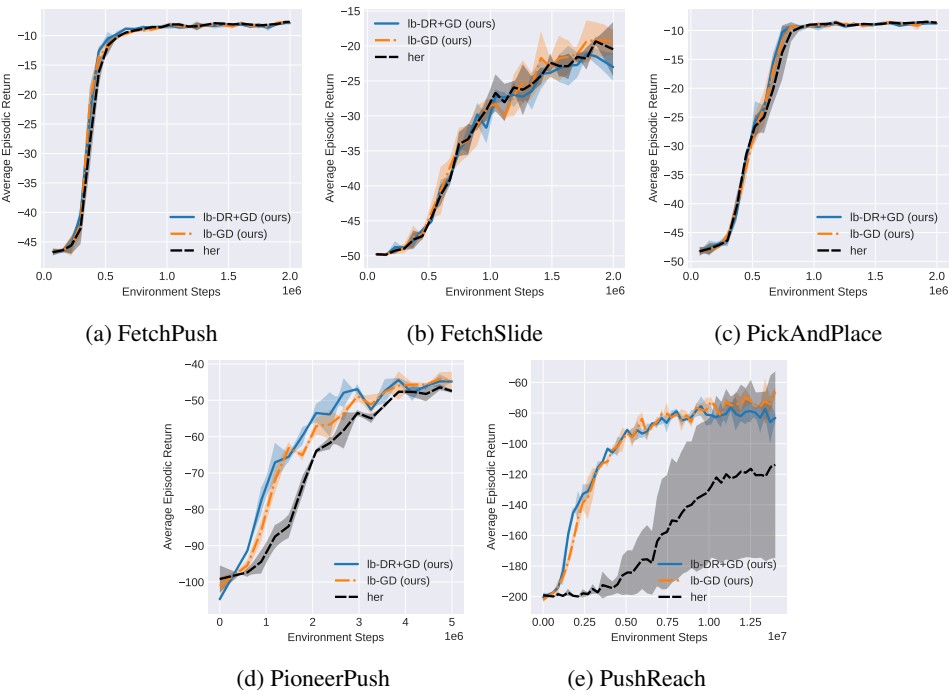

(a) FetchPush  (b) FetchSlide  (c) PickAndPlace

(d) PioneerPush  (e) PushReach

Figure 8: Value target lower bounding with goal distance return (lb-GD) and lb-DR+GD vs HER on episodic FetchEnv and Pioneer tasks. Solid curves are the mean across three seeds, and shaded areas are +/- one standard deviation.

### A.4.2    LB-GD (GOAL DISTANCE RETURN) AND LB-DR+GD VS HER

Figure 8 compares lower bounding with goal distance return (lb-GD) and lower bounding with both goal distance and discounted return combined (lb-DR+GD) against the much stronger HER baseline, on the goal conditioned episodic FetchEnv and Pioneer tasks.

On the easier FetchEnv tasks, lower bounding is similar as HER, but on the more challenging Pioneer Push and Reach tasks, lower bounding is able to achieve over 70% more sample efficiency. It seems the harder the task, the wider the margin of gain.

#### A.4.2.1    Value learning plots

This section presents plots of learned value and how often value is improved by the proposed methods, in order to show the effect of lower bounding on value improvement.

Figure 9 shows the fraction of training experience where the lb-GD is higher than the Bellman value target from HER, in the goal conditioned (episodic FetchEnv and Pioneer) tasks, and the learned value. It seems, for FetchEnv tasks, where lb-GD only performs slightly better than HER, the fraction of experience with improved value target is quite small (less than 1%). Hindsight relabeling is probably already producing fairly high value targets. For Pioneer Push and Reach tasks, lb-GD performs much better in average return, and the fraction of experience with higher value target is also much larger (peaking around 2-8%).

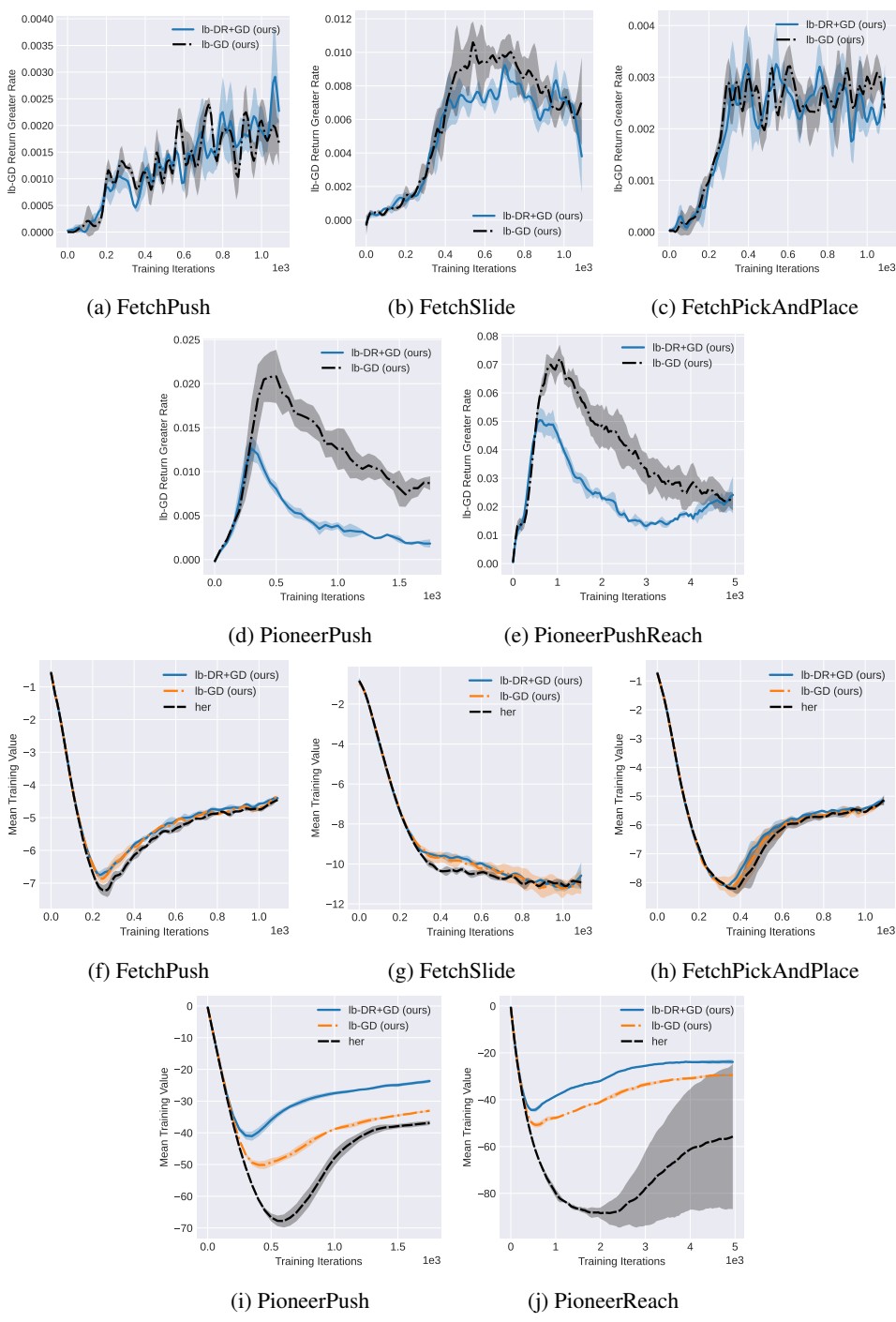

Figure 9: Fraction of training experience where lb-GD or lb-DR+GD value target is greater than the Bellman target (a-e) and learned values (f-j), on episodic FetchEnv and Pioneer tasks, plotted against the number of training iterations. Solid curves are the mean across three seeds, and shaded areas are +/- one standard deviation.

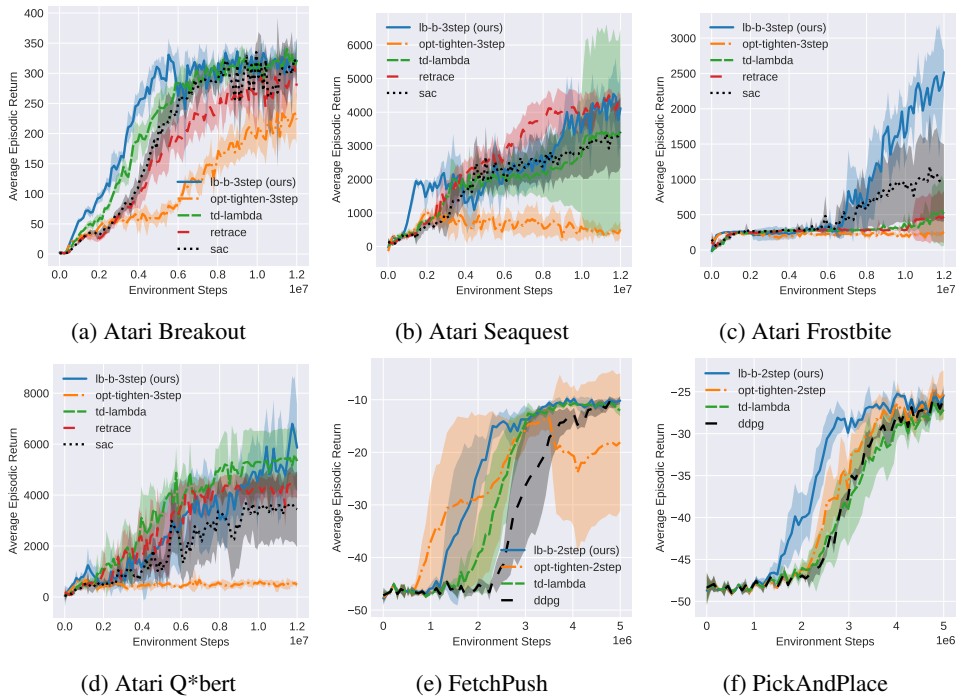

Figure 10: Evaluated average return of value target lower bounding with n-step bootstrapped return (lb-b-$n$step) vs SAC or DDPG, td-lambda, Retrace and optimality-tightening on Atari games and the original/non-episodic FetchEnv tasks. $n = 3$ for Atari and 2 for FetchEnv tasks. Solid curves are the mean across five (for Atari) or three (others) seeds; shaded areas are +/- one standard deviation.

### A.4.3 $n$-STEP BOOTSTRAPPED METHODS

Figure 10 shows performance of lb-b-$n$step methods on a subset of the Atari games (episodic) and the original (non-episodic) FetchEnv tasks. Besides SAC/DDPG, baselines also include n-step methods such as td-lambda, Retrace (Munos et al., 2016) and optimality tightening (He et al., 2017). lb-b-$n$step methods are at least as good as the best baseline method, and clearly outperforms the baselines in two of the six tasks.

#### A.4.3.1 Value learning plots

Figure 11 shows the fraction of experience where lb-b-$n$step lower bounds are above one step Bellman value targets. There seems to be a correlation between improving the value target over more experience and actually improving the policy, at least for FetchEnv tasks. The correlation does not seem as clear as that of lb-DR. With bootstrapping, the fractions are generally higher than those of lb-DR in Figure 6, potentially due to overestimated bootstrap values.

### A.4.4 $n$-STEP METHOD ABLATIONS

The following three sections include ablations which show the lower bounding methods to be robust to variations in the hyperparameters.

#### A.4.4.1 lb-b-$n$step with different $n$ (number of steps)

Figure 12 shows the effect of how the number of steps $n$ in $n$-step bootstrapped return impacts lower bounding performance. The lb-b-$n$step method is not very sensitive to the value $n$, typically, the higher the value $n$ the better the performance, while n-step methods like td-lambda or Retrace would degrade a lot as $n$ increases above 3 or 4. We also observe lower value overestimation as $n$ increases (Figure 16).

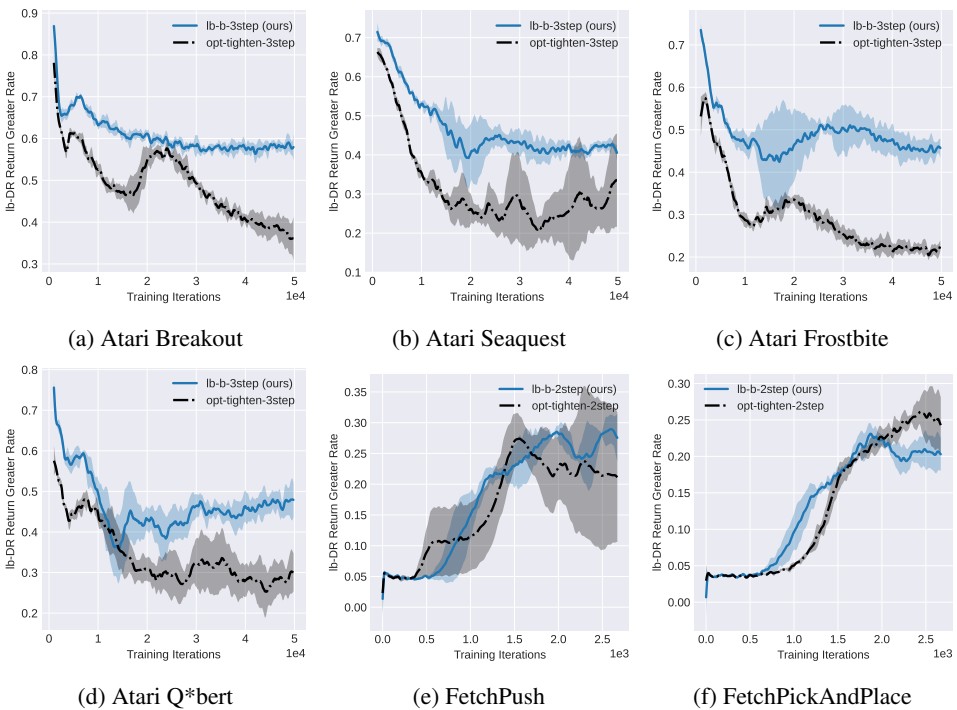

Figure 11: Fraction of training experience where lb-b-$n$step value target or optimality-tightening lower bound is greater than the value target of baselines SAC (for Atari games) and DDPG (for FetchEnv tasks), plotted against the number of training iterations. Solid curves are the mean across five (for Atari) or three (others) seeds, and shaded areas are +/- one standard deviation.

### A.4.4.2   lb-b-$n$step vs lb-b-$n$step-only (only n-th step)

Figure 13 shows the effect of taking a maximum of all 2- to n-step bootstrapped returns versus only using the n-step bootstrapped return. It seems using the maximum bootstrapped return of all 2- to n-steps, hence a tighter lower bound, works better than only using the n-step return.

### A.4.4.3   lb-b-$n$step (bootstrap) or lb-DR (episodic return)

In continuing tasks (with negative rewards), we have to use the bootstrapped lb-b-$n$step method. But for episodic tasks, should we use bootstrapped return or episodic return as value target lower bound? In theory, lb-b-$n$step-only becomes lb-DR when $n$ is large enough. In practice, in terms of effectiveness, we can compare lb-b-$n$step with lb-DR on the Atari games (Figure 10 and 5 respectively). lb-b-$n$step is better than lb-DR on Atari Breakout. On Seaquest, the two are similar. On the other two games: Frostbite and Q*bert, episodic return is better. It seems lb-DR is better on tasks where rewards are more sparse and longer term planning is needed. In terms of efficiency, as $n$ becomes larger, the memory and compute efficient lb-DR method will become more attractive. Overall, both methods show a clear advantage over the baselines.

### A.4.4.4   Value learning plots

Figure 14 shows the fraction of experience where lb-b-$n$step lower bounds are above one step Bellman value targets.

Figures 15 and 16 show the learned value of the lb-b-$n$step and lb-b-$n$step-only methods. It's convenient to look at the FetchEnv tasks which should always have non-positive value. From Figure 16(a,b), it seems as $n$ increases, value decreases, maybe due to more accurate estimates of value. From Figure 16(c,d), it seems the tighter lower bounds of lb-b-$n$step method do lead to slightly more overestimation in value.

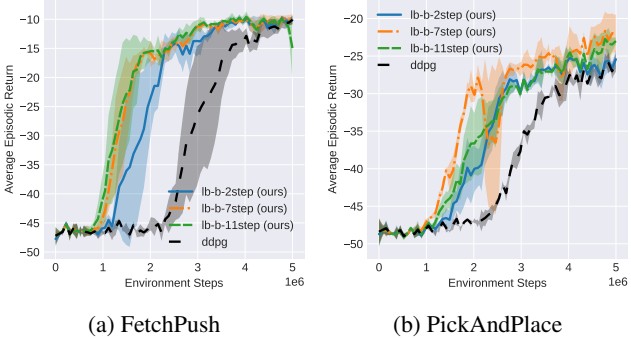

(a) FetchPush        (b) PickAndPlace

Figure 12: Evaluated average return of value target lower bounding with n-step bootstrapped return (lb-b-$n$step) with different $n$ on the original/non-episodic FetchEnv tasks. Solid curves are the mean across three seeds, and shaded areas are +/- one standard deviation.

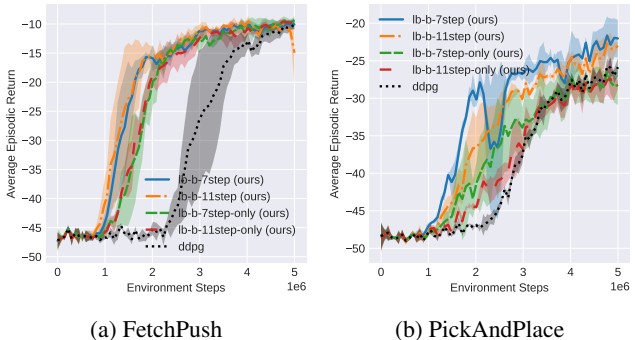

(a) FetchPush        (b) PickAndPlace

Figure 13: Evaluated average return of value target lower bounding with all n-step bootstrap (lb-b-$n$step) and nth-step only (lb-b-$n$step-only) on the original/non-episodic FetchEnv tasks. Solid curves are the mean across three seeds, and shaded areas are +/- one standard deviation.

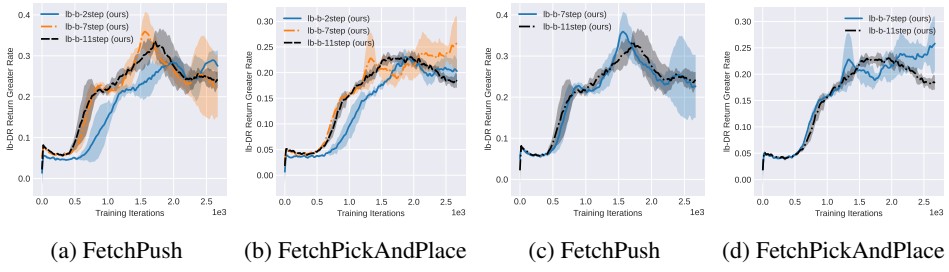

(a) FetchPush    (b) FetchPickAndPlace    (c) FetchPush    (d) FetchPickAndPlace

Figure 14: Fraction of training experience where lb-b-$n$step or lb-b-$n$step-only improves over the baseline Bellman value target, evaluated on the training experience and plotted against the number of training iterations. Solid curves are the mean across three seeds, and shaded areas are +/- one standard deviation.

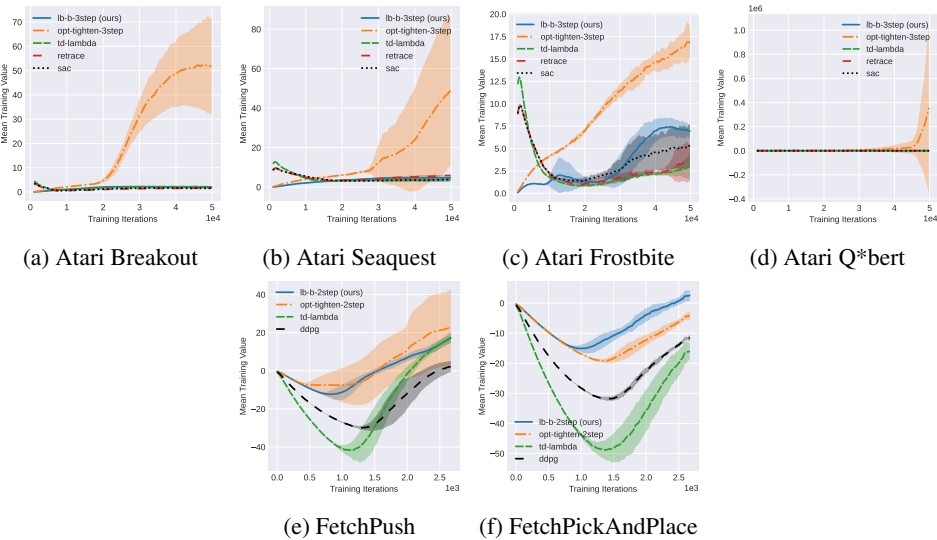

Figure 15: Learned values of lb-b-$n$step and SAC (for Atari games), DDPG (for FetchEnv tasks), optimality-tightening, td-lambda and Retrace, evaluated on the training experience and plotted against the number of training iterations. Solid curves are the mean across five (for Atari) or three (others) seeds, and shaded areas are +/- one standard deviation.

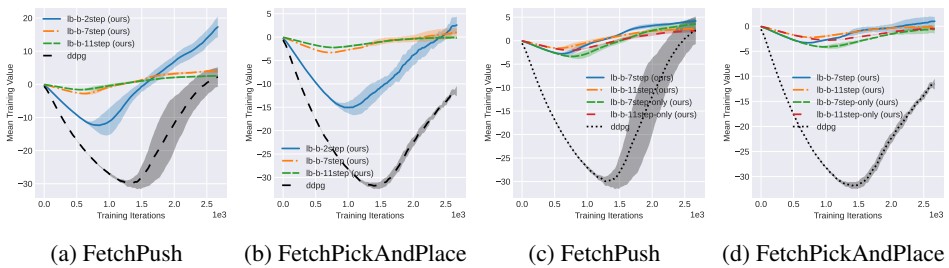

Figure 16: Learned values of lb-b-$n$step, lb-b-$n$step-only, DDPG (for FetchEnv tasks), optimality-tightening, td-lambda and Retrace, evaluated on the training experience and plotted against the number of training iterations. Solid curves are the mean across three seeds, and shaded areas are +/- one standard deviation.

## A.5  $n$-STEP RETURN BASED METHODS

### A.5.1  $n$-STEP RETURN METHODS

lb-b-$n$step methods and n-step return methods are similar in their data and computation requirements, and we already compare their performance in Figure 10.

Here, we additionally compare lb-DR directly with n-step methods in Figure 17.

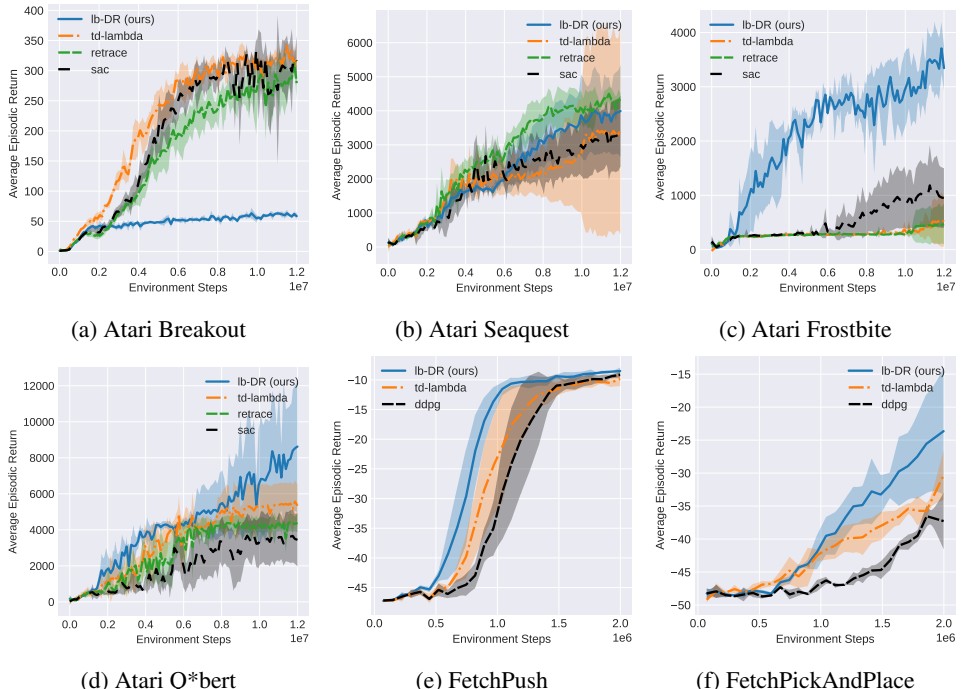

(a) Atari Breakout      (b) Atari Seaquest      (c) Atari Frostbite

(d) Atari Q*bert      (e) FetchPush      (f) FetchPickAndPlace

Figure 17: Evaluated average return of value target lower bounding with discounted return (lb-DR) vs SAC or DDPG, td-lambda and Retrace on four Atari games and the episodic FetchEnv tasks. Solid curves are the mean across five (for Atari) or three (others) seeds, and shaded areas are +/- one standard deviation.

During experimentation, we found that n-step methods are typically harder to tune and more expensive to compute.

1) Tuning $n$: A small $n$ for n-step methods works similarly as the baseline one-step method, and a larger n hurts performance. This is likely due to the off-policy bias in n-step return causing the n-step estimate to be potentially worse than the one-step estimate. Introducing importance sampling weights (Retrace) would help reduce the bias, but at the same time significantly downweight the off-policy high return experiences, making an ineffective use of such experiences.

None of these issues exist in value target lower bounding: (a) It does not incur any off-policy bias, and (b) as long as an experience renders high reward, being off-policy does not affect its ability to improve the value target.

2) Tuning involves other hyperparameters like the td-lambda parameter, replay buffer size, prioritized replay (to potentially expire old experiences and sample recent ones more frequently), target network update parameters (to reduce potential overestimation), and parameters for importance sampling. But still, after all the tuning, it only slightly outperforms one-step DDPG on FetchEnv or SAC on Atari games, and is often below the lower bounding methods. For td-lambda and Retrace, the best performance comes from 2-step (for FetchEnv) or 3-step (for Atari) td with $\lambda = 0.95$, all other parameters the same as the baseline DDPG or SAC. Retrace underperforming the baseline in Breakout and Frostbite is similarly observed in the original paper (Munos et al., 2016).

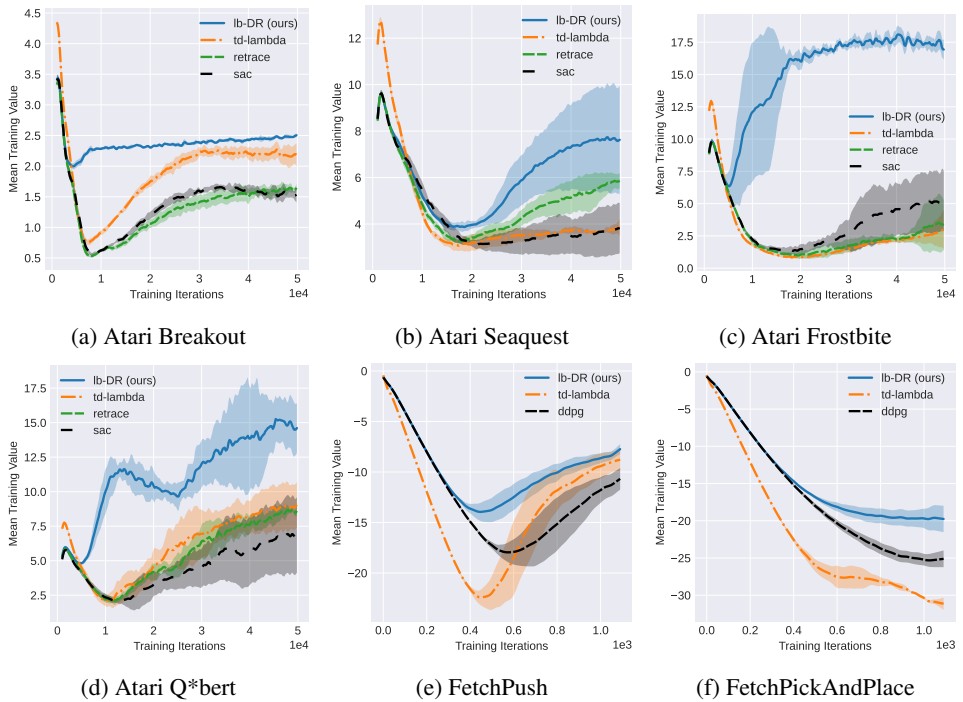

Figure 18: Learned values of lb-DR and SAC (for Atari games), DDPG (for FetchEnv tasks), td-lambda and Retrace, evaluated on the training experience and plotted against the number of training iterations. Solid curves are the mean across five (for Atari) or three (others) seeds, and shaded areas are +/- one standard deviation.

On the other hand, value target lower bounding with episodic return requires no hyperparameter tuning, (lb-b-$n$step with bootstrapping is not very sensitive to the choice of $n$), and learns faster on most tasks and converges higher on many tasks.

3) Computing n-step td-lambda return requires more computation due to evaluating value networks on all n-steps of the experience. It limits how large $n$ can be due to GPU memory limits, and slows down training time significantly with a large $n$.

On the other hand, value target lower bounding precomputes and stores episodic discounted return in the replay buffer, and incurs very little additional computation.

Overall, n-step methods are much more expensive and difficult to use, and the much simpler and effective lower bounding methods maintain an advantage in both effectiveness and efficiency. We show the performance comparison in Figure 17 with learned values in Figure 18.

### A.5.2 OPTIMALITY TIGHTENING WITH N-STEP RETURNS

He et al. (2017) use bootstrapped n-step return to lower and upper bound the value during training. They frame the problem as a constrained optimization problem where the distance between the value and the Bellman value target is minimized subject to the constraints that the value function must be within the lower (and upper) bounds. This is similar to our lb-b-$n$step method, except instead of applying a constraint on the value, we use the bootstrapped return to directly lower bound and improve the value target, which is likely more optimal and more efficient. In their experiments, the Lagrangian multiplier was fixed, which would likely lead to suboptimal solutions, and no theoretical guarantee was given. For episodic tasks, even more efficient and effective methods like lb-DR exist.

Some detailed differences:

1) The prior work bounds the value function itself (similar to lower bound q learning (Oh et al., 2018; Tang, 2020)), instead of bounding the Bellman value target. This could cause suboptimal training

because the Bellman target itself could be outside the bounds, causing contradictory training targets and losses. Imagine the current value for a state is 1, its Bellman value target may be a low 0, and the lower bound may be a high 2, then it's unclear which way the value function should go. It will depend largely on the mixing weight between the two losses $\lambda$ and whether initial values overestimate, which can be hard to tune in practice. In their experiments, a fixed Lagrangian multiplier $\lambda$ was used, which makes the method likely non-optimal.

2) In order to compute the bootstrapped values, the value network needs to be evaluated on all n future time steps, severely increasing GPU memory consumption and compute. Because of this increase in compute, in experiments, it could only look at a limited (4) timesteps into the future, while lb-DR and lb-b-$n$step-only can look ahead much further with very little extra computation and storage.

We implemented the optimality tightening method (He et al., 2017), using only the lower bounds to be comparable to the other methods, and integrated it into our baselines. We ran on FetchPush and FetchPickAndPlace with hyperparameters number of time steps $n = 2$, and the penalty coefficient $\lambda = 4$ following the original paper. Results in Figure 10 show optimality tightening to be either not as stable or not as optimal as our lb-b-$n$step. We also ran it on Atari games, and found that optimality tightening overestimates value a lot, leading to much worse behavior than even the SAC baseline. The constrained optimization formulation might have an adverse effect during RL training, and the upper bounds in the original optimality tightening work may be necessary to bring training back on track.

It will be interesting to see whether value target upper bounding with bootstrapped values (as proposed in (He et al., 2017) as value upper bound constraints) can lead to optimal converged value in theory, and whether the upper bounds would help optimality tightening in experiments. However, tight upper bounds are not as readily available as lower bounds. This is because during RL training, the agent's performance hopefully increases, which makes it difficult to come up with a proper upper bound before the optimal value is known. Additionally, the motivation for using upper bounding would be, for example, to avoid value overestimation, which is very different from the motivation of faster value improvement for lower bounding. Hence, we see upper bounding as out of the scope of this work.

## A.6 VALUE TARGET LOWER BOUNDING ON DDQN

Because DQN is a more popular baseline for the Atari games, we've also applied value target lower bounding (lb-DR) on DDQN, and ran on a subset of the Atari games. Different from (van Hasselt et al., 2015), our implementation of DDQN finds the maximal action using the target critic network and evaluates the target value on the target network. It also uses two critic replicas like in (Fujimoto et al., 2018). Figure 19 shows the results. Compared to SAC (Figure 5), DDQN either lowers the baseline performance or the treatment (lb-DR) performance, and does not seem as strong as SAC.

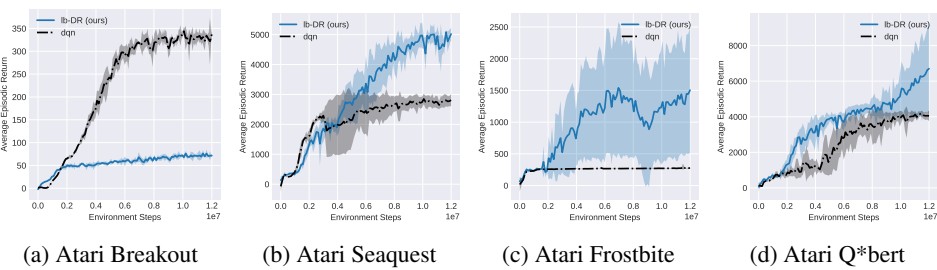

| (a) Atari Breakout | (b) Atari Seaquest | (c) Atari Frostbite | (d) Atari Q*bert |

Figure 19: Evaluated average return of value target lower bounding with discounted return (lb-DR) implemented on DDQN vs DDQN baseline on four of the Atari games. Solid curves are the mean across five seeds, and shaded areas are +/- one standard deviation.

A.7 UNBIASED EXPERIENCE REPLAY

RL methods like Hindsight Experience Replay can introduce a hindsight bias (Plappert et al., 2018; Lanka & Wu, 2018; Blier & Ollivier, 2021) in training due to a future step from an experience being used to relabel an earlier step of the same experience, possibly biasing the distribution of training experience, overestimating value and making the task seem easier than it really is. MuZero (Schrittwieser et al., 2019) assumes deterministic transitions and could also exhibit bias in stochastic environments (Antonoglou et al., 2022). Similar to HER, in this work, the future return from an experience when directly used to lower bound the value target computed from the same experience could also introduce bias to the value target. This bias could lead to value overestimation, as shown in the example below.

Assume state $S_0$ always goes to $S_1$. $S_1$ gives reward $+2$ or $-2$, 50% of the times randomly, and transitions to the terminal state. For this case, the true values are $v(S_0) = v(S_1) = 0$. For one lucky experience with reward $+2$, empirical return for $S_0$ is $2\gamma$, and for an unlucky one, $-2\gamma$. Averaging the two leads to the unbiased estimate of return for $S_0$, 0, which could be safely used as a valid lower bound. However, if the lower bound is computed only using one lucky experience or one unlucky experience, then the lower bounded value target for $S_0$ is $\gamma \max(2, v(S_1))$ and $\gamma \max(-2, v(S_1))$ respectively, averaging to be $\frac{\gamma \max(2,v(S_1))+\gamma \max(-2,v(S_1))}{2} = \frac{2\gamma+0}{2} = \gamma$, which overestimates $v(S_0)$.[4]

Existing work ARCHER (Lanka & Wu, 2018) and stochastic MuZero (Antonoglou et al., 2022) already follow up HER and MuZero, proposing unbiased sampling or model enhancements to address the bias. Thus, the specific way of using the episodic or bootstrapped return from the sampled transition to lower bound the value target of the transition could be biased in stochastic environments, but future work may propose unbiased sampling methods to reduce or remove the bias of empirical return based lower bounds.

Luckily, when reward sequences are reproducible, which is largely satisfied for many tasks in the macroscopic physical world, where reproducibility is often assumed, empirical return lower bounding always produces an unbiased estimate of the value target. For the tasks in this paper, both Atari games and physical simulators have near deterministic transitions, with pseudo randomness affecting reproducibility. However, it seems the impact is small, as we do not observe any apparent overestimation, and lower bounding still produces quite large improvements. This is consistent with the observations of prior works such as Fujita et al. (2020).

A.8 MORE ABOUT EPISODIC CONTROL

The original episodic control (Blundell et al., 2016) simply applies $Q(s, a) = \max(Q(s, a), R)$ where $R$ is the newly observed return for the entry $(s, a)$ in the episodic memory $Q$. This lower bounds the Q value directly, and if Q is already overestimated due to initialization or function approximation, it will stay overestimated, because the overestimation never goes through Bellman contraction.

Hu et al. (2021) corrects the problem by using implicit memory based planning in the following way. For an episode of length $T$, current step $t \leq T$, the value target is defined recursively as

$$R_t = \begin{cases} r_t + \gamma \max(R_{t+1}, Q_\theta(s_{t+1}, a_{t+1})) & \text{if } t < T \\ r_t & \text{if } t = T \end{cases} \quad (12)$$

This back propagation process can be further unrolled as follows. Define value as

$$V_{t,h} = \begin{cases} r_t + \gamma V_{t+1,h-1} & \text{if } h > 0 \\ Q_\theta(s_t, a_t) & \text{if } h = 0 \end{cases} \quad (13)$$

Given planning horizon $h$, the value target for the value $Q_\theta(s_t, a_t)$ is just $R_t$:

---

[4]For this argument, we can simply assume $v(S_1)$ to be 0, the correct value.

$$R_t = V_{t,h^*}, \text{ where } h^* = \underset{h>0}{\arg\max} V_{t,h} \tag{14}$$

The maximum over planning horizons $h$ in Equation 14 allows existing values from the episodic memory $Q_\theta(s_i, a_i)$ for $t < i \leq T$ to influence and lower bound the return of the current trajectory. As the episodic memory contains information about the trajectories from past experience, the final value target is influenced not just by the current trajectory, but also past trajectories that share a similar state as any of the states on the current trajectory.

$Q_\theta(s_t, a_t)$ and $V_{t,h}$ are carefully defined to be 0 for all states after the episode ends i.e. $t > T$. This allows the episode end to always have the correct value to begin with, so proper value can be backed up the trajectory to all earlier states. Equation 14 also importantly requires $h$ to be above 0 when taking the maximum over planning horizons. This way, the value target for $Q_\theta(s_t, a_t)$ is not influenced by $Q_\theta(s_t, a_t)$ itself. Only $Q_\theta(s_{t+i}, a_{t+i})$ (for $i > 0$) can influence the value target for $Q_\theta(s_t, a_t)$, and they have to go through the Bellman operator (see Equation 13 when $h > 0$). This means any initial overestimation will contract.

Episodic memory provides an interesting tool to compute tight value target lower bounds for near deterministic tasks, with a computational cost similar to a typical $n$-step return method.

## A.9 ADDITIONAL RELATED WORKS

Planning methods can look into the future to achieve higher value targets and better control. Examples include Monte Carlo Tree Search (MCTS) (Schrittwieser et al., 2019; Ye et al., 2021) and Model Predictive Control (MPC) or receding horizon planning with raw actions (Chua et al., 2018; Hafner et al., 2019; Zhang et al., 2022), options (Silver & Ciosek, 2012), or subgoals (Nasiriany et al., 2019; Nair & Finn, 2020; Chane-Sane et al., 2021). Planning methods use either a dynamics model together with the learned value or just the learned value (in the case of goal conditioned tasks) (Nasiriany et al., 2019) to improve policy or value estimates. Planning typically happens during roll out (Nasiriany et al., 2019), but can also be used to improve the value target, as in Reanalyze of MuZero (Schrittwieser et al., 2019; Ye et al., 2021). During value improvement, if planning takes the maximum over a set of possible future values (e.g. from different trajectories as in the case of MPC), and if this set includes the one step Bellman value target, then the planner is essentially using alternative trajectories and their values to lower bound the Bellman value target. In this sense, the theory developed here can potentially justify and improve Reanalyze. In general, planning is orthogonal to value target lower bounding, and typically requires additional components and a lot more compute than the basic TD learning does. Therefore, we leave it to future work to explore the synergy between the two.

Our work can be seen as an extension of the admissible heuristics (bounds) method (Russell & Norvig, 2020) into the RL domain. Admissible heuristics speed up the search for the optimal solution. For example, it is common practice to lower and upper bound the returns to the possible region, e.g. Andrychowicz et al. (2017) bounds value targets between $[-\frac{1}{1-\gamma}, 0]$. Episodic return (Section 3.1) provides a tighter bound. In addition, our work extends the idea to the case of lower bounding with bootstrapped values, which can overestimate the value. We prove it still converges to the optimal value.

Kumar et al. (2020) (DisCor) also recognizes that bootstrapped value targets can be inaccurate. This bias impacts learning adversely under function approximation. DisCor uses distribution correction to sample experience with accurate bootstrap targets more frequently, while value target lower bounding aims to directly reduce the bias.

## A.10 POTENTIAL IMPROVEMENT

Note that the goal distance based return (lb-GD) of Section 3.1.1 is a very simple way of arriving at a reasonable lower bound with near zero additional computation. The bound could be made tighter. Typically, an $L_2$ distance threshold is used to judge goal achievement, which will likely be satisfied a few time steps before exactly arriving at the hindsight goal. To compute such a tighter bound would require evaluating the reward function across the trajectories of experience using all possible hindsight goal states, and storing them in the replay buffer, i.e. episode length squared more

computation and more storage space. It may be worth doing when episodes are short, or doing it only for a small number of time steps into the future when e.g. rewards are non-negative.

