# OpenReview forum: "Faster Reinforcement Learning with Value Target Lower Bounding"
_ICLR.cc/2023/Conference — Submitted to ICLR 2023_

### Official Review · Reviewer_JN96 · 2022-10-21

**Confidence:** 4
**Correctness:** 4
**Technical Novelty And Significance:** 2
**Empirical Novelty And Significance:** Not applicable
**Recommendation:** 5

**Clarity, Quality, Novelty And Reproducibility:**

## Clarity
- The paper reads well and is easy to follow.

## Quality
- The paper is written in a rather informal manner and I believe some sentences/paragraphs need to be polished. A few examples are:
    - Section 1 ends abruptly. It would be better to move the last paragraph to another section (or merge it with the previous one) and introduce a brief overview of the results.
    - (Sub)section titles are not capitalized in a consistent way (e.g., Value target lower bounding vs An Illustrative Example).
    - Section 6, "It is quite easy to introduce biases and inefficiencies into the process and end up with a suboptimal or inefficient algorithm.": This requires supporting evidence.
    - Section 6, "In the experiments, the Lagrangian multiplier is fixed rather than being learnt, which would likely lead to suboptimal solutions.", "We lower bound the value target directly, which is simpler, more efficient, and likely more optimal.": These require supporting evidence too.
    - The paper contains a dummy acknowledgment section.

## Novelty
- The proposed value target lower bounding technique (lb-DR) is previously used by Fujita et al., as acknowledged by the authors. Nevertheless, the paper still provides a generalization of this technique and justifies it theoretically.
- While the theoretical results in the paper seem novel (and correct), I am not familiar with the literature in this area and am not completely certain about its theoretical novelty.

## Reproducibility
- The code is not released.

## Minor issues
- Appendix A.3: Netowrk -> Network

**Strength And Weaknesses:**

## Strengths
- The proposed method is simple and sensible.
- Their theorems provide insights into their method.
- The limitations of the proposed method and theorems are well addressed throughout the paper.

## Weaknesses
- The main weakness of this work is on its experimental setup. They use somewhat unusual combinations of base RL algorithms and environments. For example, they use SAC for Atari games, which is an uncommon choice especially given that Atari games have a discrete action space. I believe it would be desirable to test and compare their method with more widely used setups. How does the method perform with DQN or Rainbow on Atari games? Also, the authors only evaluate their method on an unusual subset of 17 games. Is there any previous work that uses the same setting? Could the authors provide additional results on the other games?
- The novelty of the method is limited (please see the section below for details).
- The proposed value target lower bounding technique can only be applied to deterministic environments. That being said, I believe this technique can still benefit some (deterministic or stochastic) environments in the real world, especially when initial target values are not informative enough.
- The code is not released.

## Questions
- How do the authors modify SAC to deal with discrete actions?

**Summary Of The Paper:**

The paper suggests using a function that lower-bounds the optimal Q function (such as Monte-Carlo returns) as target values in Q-learning methods for faster convergence. The authors provide a theoretical justification for this technique and propose several variants for episodic/goal-reaching/non-episodic settings. They demonstrate that the proposed technique applied to Q-learning methods (SAC, DDPG, TD3) helps improve performance in diverse environments.

**Summary Of The Review:**

I believe this is a borderline paper. Since most of my concerns lie in the experimental results, I would be happy to increase my score if the authors provide a more extensive evaluation of their method.

*Post-rebuttal update*: Thank you for the response. I checked the comparisons between lb-DR and DDQN on four Atari games in Appendix A.6, but I am still not fully convinced regarding lb-DR's performance as the number of games is highly limited (4 out of 56 games for DDQN and (an unusual set of) 17 games out of 56 for SAC), especially given the simplicity of the method. Hence, I would vote for a weak reject for now.

---

> ### Author Response · Authors · 2022-11-15
> **Response to reviewer JN96**
>
> Thank you for the careful review and detailed feedback.
>
> Re: Novelty
>
> We'd like to point out that we provide a generalization of the earlier method (Fujita et al. 2020) to the non-episodic case, using bootstrapped return, in addition to the results with the episodic case.  Our theory also covers both the episodic and the non-episodic cases, and we experimented with both cases.  To the best of our knowledge, this general framework, theory and experiments with lower bounding with bootstrapped return are novel.
>
>
> Re: Experimental setup
>
> We've explained why we use SAC over DDQN in the general response (we did experiment with and include results on DDQN).  Rainbow is a rather complex method to compare to, but we did include n-step td methods as baselines (Appendix A.4.3 and A.5).  N-step return contributed significantly to Rainbow's success, but there is still benefit in using episodic return lower bounding, likely because episodic return looks much further ahead than typical n-step methods could afford (due to computational cost).
>
>
> Re: Using a subset of Atari games
>
> Atari is such a classic benchmark that it is very tempting to include all the games, especially if the goal were to top the benchmarks.  The considerations for the experiments in this paper are different.  We only want to demonstrate how the lower bounding method works on a wide range of environments against strong baselines.  That’s why for Atari we include some classic games commonly used in prior works and additionally randomly select a subset.  We hope this includes an enough number of games with enough variety, but also reduces unnecessary computation (energy use and pollution), especially given that the current set of games already shows statistically significant gains, and supports our claim.  (Appendix A.2.1 has more details about how the Atari games were sampled.)
>
>
> Re: SAC on discrete actions
>
> Thanks for pointing this out.  We’ve added the following to the Appendix A.3:
>
> To use SAC on discrete actions, we simply swap out SAC’s value networks (which take in continuous action and state as input and produce the value) with DQN’s Q networks (which take in state as input and output $n$ heads, each head for one action logit).  We pass the output of the Q networks (administered by the temperature $\alpha$) through softmax to compute the action probabilities, and to further sample actions.  We ignore actor losses because there is no separate actor network, and only use critic losses to train the Q networks.
>
> The reason we can use Q with temperature and softmax to compute action probabilities is that for discrete actions, maximizing the SAC objective $\max_{p}{[p(a_i, s) (Q(s, a_i) - \alpha \log{p(a_i, s)})]}$ subject to $\sum_i{p(a_i, s)} = 1$ directly gives the solution $p(a_i, s) \propto \exp(Q(s, a_i) / \alpha)$.  See Appendix D of [Yu et al, 2021] (TAAC: Temporally Abstract Actor-Critic for Continuous Control) for a similar derivation.
>
>
> Re: code release
>
> Our code is ready for release, but the specific URL is withheld for anonymity during review.  We plan to release the code upon publication as mentioned in the reproducibility statement.
>
>
> Re: quality
>
> Thanks for the detailed feedback about quality.  Changes made:
>
> * Included a summary of results at the last part of Section 1.
> * Made capitalization of subtitles consistent.
> * Added supporting evidence to the claim about "easy to introduce biases" (Section 6).
> * The optimality tightening paper itself mentions this suboptimality for not tuning the Lagrangian multiplier.
> * Removed the dummy Acknowledgement section.

---

### Official Review · Reviewer_8emL · 2022-10-24

**Confidence:** 3
**Correctness:** 4
**Technical Novelty And Significance:** 2
**Empirical Novelty And Significance:** Not applicable
**Recommendation:** 6

**Clarity, Quality, Novelty And Reproducibility:**

-

**Strength And Weaknesses:**

Strengths:

- The paper is thorough in that the authors do highlights subtle differences between convergence and contractions, and I do agree with authors that we can generally ignore that fact for empirical studies as we apply this for multiple amounts of time.

- This also builds into and connects to a lot of episodic learning papers where it can be viewed as simply adding a lower bound term into the value learning and get better rewards[1,2,3], It would be wonderful if the authors could throw some light towards this as these advances have mostly been without theoretical insights.

- The authors provide reasoning for discrepancies in the results such as “breakout” where the clipping of rewards has a significant effect in the lower bounds.


Weakness:

- It is not clear on this method will improve sample efficiency for algorithms that are designed to be sample efficient to begin with, for example rainbow. Will simply increasing the number of backups during training iterations have a similar effect?


Minor clarity

- When we say ““One such condition is when reward sequences are reproducible, under which, the empirical return from a single experience provides an unbiased estimate of the lower bounded value target” ([pdf](zotero://open-pdf/library/items/4UI97JPI?page=31)) ” Does it mean that it is stochastic in nature ? [pg 31]

- Could we reprhase “upper bounds are not as readily available as lower bounds” to “tight upper bounds are not as readily available as tight lower bounds”. As v_max can be a trivial upper bound. [pg 30]


[1] Ma, X., Yang, Y., Hu, H., Liu, Q., Yang, J., Zhang, C., Zhao, Q., & Liang, B. (2022). Offline Reinforcement Learning with Value-based Episodic Memory. *ArXiv, abs/2110.09796*.

[2] Sarrico, M., Arulkumaran, K., Agostinelli, A., Richemond, P.H., & Bharath, A.A. (2019). Sample-Efficient Reinforcement Learning with Maximum Entropy Mellowmax Episodic Control. *ArXiv, abs/1911.09615*.

[3] Lin, Z., Zhao, T., Yang, G., & Zhang, L. (2018). Episodic Memory Deep Q-Networks. *ArXiv, abs/1805.07603*.

**Summary Of The Paper:**

The paper explores the use of lower bounds for for value targets in order to improve the sample efficient in Reinforcement Learning algorithms. The authors outline the design of easily computable value lower bounds across different environment settings (episodic or continuous) and algorithms (n-step TD, SAC, HER). They theoretically justify the use of these lower bounds for faster convergence of value iteration. The intuition behind this is closely related to seeding the initial value to a lower bound during value iteration in tabular settings for faster convergence. Moreover, the empirical results demonstrate that the use of value lower bounds under different settings does improve the sample efficiency of these algorithms.

**Summary Of The Review:**



Overall, the paper builds on the rich body of work done previously with different algorithms and the experiments shown are comprehensive and show promising results. However, it further experiments with more sample efficient algorithms such as rainbow and its variants is most welcome.

---

> ### Author Response · Authors · 2022-11-12
> **Response to Reviewer 8emL**
>
> Thank you for the careful review and suggestions.
>
> Re: episodic control
>
> Thanks for pointing us to this line of research.  They are very relevant.  We included the following discussion in the updated draft:
>
> Episodic control (Blundell et al., 2016) and follow-up works (Lin et al., 2018; Sarrico et al., 2019; Hu et al., 2021; Ma et al., 2022) use episodic memory from past experiences to develop value lower bounds.  When multiple experiences start from the same state, the maximum episodic return is stored in the episodic memory slot for that state.  This provides a potentially tighter lower bound than the episodic return used in this paper (which only uses the episodic return from the one episode where the transition is sampled).  During control, the action that maximizes the stored episodic return is picked.  For the general non-tabular case, function approximation is used to represent states, which bootstraps values of states never encountered in training before.  The original episodic control and a few of the variations (Blundell et al., 2016; Lin et al., 2018; Sarrico et al., 2019) lower bound the value function itself, not the value target, similar to how Self Imitation Learning (SIL) does (Oh et al., 2018).  Thus an initially overestimated episodic value, due to either improper initialization or function approximation, stays overestimated throughout training, because it never goes through the Bellman operator.  These methods may not converge to the optimal value even for tabular deterministic environments.  Follow-up works (Hu et al., 2021; Ma et al., 2022) use implicit memory based planning, which essentially lower bound the value target with a function based on the episodic return, and avoid the earlier overestimation problem (see Appendix A.8 for details).  Maximum entropy Mellowmax episodic control (Sarrico et al., 2019) uses a temperature controlled softmax based operator to generate the action probabilities and is similar to the discrete version of SAC.  Overall, episodic memory is an interesting tool to come up with tight value target lower bounds, while the value target lower bounding we propose is more general along two directions: the theory works for stochastic environments, and for non-episodic tasks (with experiments in Appendix A.4.3).
>
>
> Re: increasing the number of backups during training
>
> n-step td methods allow value to be backed up more than one step at a time, and is an important part of the success of Rainbow.  We did include n-step baselines in our experiments (Appendix A.4.3 and A.5).  With a large $n$, performance of n-step methods degrades (reported by e.g. Rainbow), likely due to off-policyness (see Sutton and Barto 2018, Chap 7).  Our method, episodic return lower bounding, does not suffer from off-policyness, and can look much further ahead than typical n-step methods could.  Experiments do show additional benefit of our method even on top of n-step methods.  Appendix A.5 has more discussion.
>
>
> Responses to minor clarity:
>
> a) Re: unbiased estimates of value target and empirical return [pg 31]
>
> The training process samples batches of experience to compute lower bounds and value targets, and is stochastic in nature.  However, as long as the reward sequences in the experiences are reproducible, empirical return can be used to produce unbiased estimates of value targets.
>
> b) "upper bounds" => "tight upper bounds" is a good suggestion.  Done.

---

### Official Review · Reviewer_7q5U · 2022-10-31

**Confidence:** 3
**Correctness:** 3
**Technical Novelty And Significance:** 2
**Empirical Novelty And Significance:** Not applicable
**Recommendation:** 5

**Clarity, Quality, Novelty And Reproducibility:**

See above, paper is clearly written for the most part with some minor issues (e.g. Section 4.3), I do not see any major reproducibility issues.

**Strength And Weaknesses:**

Strength:
- The proposed method is simple and straightforward and can potentially be augmented to many RL algorithms
- The theoretical analysis does appear to be correct and the convergence result is quite intuitive
- The related work section is well-written. Since many similar but not entirely identical methods have been proposed, I think the authors did a good job of placing their work within the context of other literature and differentiating their algorithm

Weakness:
- The authors claimed to introduce a faster technique, however the only analysis on how the proposed method could be “faster” is the third to last paragraph where the authors claimed that the new algorithm is at least as fast as the original. If faster RL is one of the claims of the algorithm, some new analysis in terms of convergence rate would be much appreciated.
- I could be missing something here, in Section 3 you mentioned using the observed discounted return as a lower bound. When the environment is deterministic, this is certainly true, however in a stochastic environment, this would no longer be the case since it is always possible to get a very large return on an arbitrary rollout. Similarly, the lower bounds proposed in Section 3.2 has a similar issue.
- The illustrative example in Section 4.3 is not entirely clear, could you give some additional clarification on why the proposed method speeds up learning in this case?
- Though variations of both SAC and DDPG work for discrete action spaces, both were originally designed for continuous action spaces and are not standard for Atari. I understand the choice of SAC and DDPG since the purpose is to show improvements on a TD-based algorithm but was there a particular reason for using the Atari environments here instead of e.g. MuJoCo?
- To add on to the previous point, I think while the experiments do show that the proposed methodology can work on complex difficult tasks, it doesn’t help us understand where the improvements actually come from. In this regard, a toy example to demonstrate for example under what scenarios we see faster convergence would be extremely helpful.
- 3-5 seeds seem very small to give any conclusive results though I understand in some labs computational resources could be limited.


**Summary Of The Paper:**

This paper replaces the value target in TD-based methods with a lower bound of the optimal value function. Convergence was shown in the tabular case and the authors introduced several lower bounds that can be applied to practical algorithms. The authors demonstrated the effectiveness of their method on a series of challenging high-dimensional tasks.

**Summary Of The Review:**

While I think the paper does have some merits and introduces a simple yet effective idea, I do not believe the paper is ready for publication in its current form.

---

> ### Author Response · Authors · 2022-11-15
> **Response to Reviewer 7q5U**
>
> Thanks for the careful and detailed review.
>
> Re: noise in experiments and more seeds
>
> Thanks again for pointing it out.  It was a mistake to use standard deviation for a non-normally distributed set of data.  This exaggerates the noise level.  New plots show that most of the times, the improvements are statistically significant.
>
> Re: Faster training
>
> By faster, we mean empirical performance, and how the proof demonstrates the conditions for faster convergence, i.e. when the lower bound function exceeds the Bellman value target.  We've also included a fairly general case as toy example (see general response), showing exactly when learning is faster.  Based on our proof so far, only some of the training iterations may benefit from a higher rate of convergence, and it depends on the quality of the lower bound function in use.  We'll keep thinking about how to demonstrate a more consistently improved rate of convergence.
>
>
> Re: benchmarking with MuJoCo simulated environments
>
> The FetchEnv we used is based on MuJoCo, which includes Fetch Push, Slide and PickAndPlace tasks.  We did not experiment with the dense reward tasks such as walker, hopper, and runner etc..  That’s because value target lower bounding efficiently propagates future rewards backward, and is a natural fit for sparse reward tasks.  (This is also explained in Section 5 first paragraph.)
>
>
> All other points about benchmarks, baselines and toy example are addressed in the general response.

---

### Author Response · Authors · 2022-11-15
**General response to reviewers' comments**

Thank you, all reviewers, for the careful review, and the helpful and detailed feedback.

Below, we list responses to the more general points shared among reviewers.  All other responses are included in the individual replies to each reviewer.


Re: the noisy Atari games plot (Figure 2a) and using more seeds

Thanks for pointing out that the Atari experiments seem noisy.  It is strange that, given the fairly consistent gains for each game (Appendix A.4), the aggregated plot (Figure 2a) still appears noisy.  Turns out standard deviation is not the right metric to use to measure the noise level of the fraction of times when treatment is above baseline.  This fraction is calculated based on binary data points of 0’s -- when treatment is not above baseline, and 1’s -- when treatment is above, which is not normally distributed.  Thus, the standard deviation of the fraction exaggerates the underlying noise level.  The right tool to use is the sign test.  (The reason for using binary decisions and the sign test to compare methods is that the scores of the Atari games vary wildly across games, and cannot be easily combined without bias to certain games, even with the human normalized scores.  The sign test ignores the magnitude of change, thus is less sensitive, but at the same time more robust, because it does not rely on any assumption about the distribution of the magnitude of the differences.  To see the magnitude of the changes, Appendix A.4 plots the game scores for each game individually.)

We’ve updated all plots in Figure 2 to show, instead of standard deviation, a reference line, above which the difference is statistically significant by the sign test (one-tailed, 0.05 significance level).  The new plots show quite clearly that during most of the training process, the treatment method is statistically significantly above the baseline.  Given the new plots, it seems five seeds is enough for Atari.


Re: illustrative example and simpler toy example

We’ve added the following simpler and more concrete example to the draft:

At the core of the illustration in Section 4.3 is this simpler example:
Consider a simple chain of $n$ states $s_1$ to $s_n$, with $s_i$ going to $s_{i+1}$.  It emits reward 1 and terminates when the agent is already at the end of the chain, state $s_n$, and emits reward 0 elsewhere.  Suppose we have already collected an experience of traversing the chain once, and start training with zero initial value.  The basic value iteration algorithm would back up the reward just one step along the chain for each iteration of training, converging in $n$ iterations.  However, value target lower bounding trains much faster.  Given the collected experience, we precompute the empirical return of each state $G(s_i) = \gamma^{n - i}$, which is also the optimal value.  Thus, one training iteration already populates all states with the optimal value.


Re: SAC as baseline for Atari

The reason for using SAC on Atari games is that the original DDQN setup with one environment for training runs very slowly.  We tuned it and during development found DDQN to generally perform worse than SAC (reported in Appendix A.3 and A.6).  Experiments on FetchEnv follow existing work closely.  We've added details about the discrete SAC implementation in Appendix A.3.


Re: empirical return lower bounding and environment stochasticity

The reviewers are mostly correct.  It is quite obvious that empirical return lower bounds the optimal value in deterministic environments.  (It could bias and overestimate the value target in stochastic environments.)  It may be less obvious, but several popular RL methods such as HER and MuZero similarly exhibit bias in stochastic environments.  (Follow up works ARCHER [Lanka and Wu 2018] and Stochastic MuZero [Antonoglou et al 2022] propose unbiased sampling methods or model modifications for stochastic environments.)  Given that the macroscopic physical world behaves near-deterministically, these prior methods are still widely adopted.  For this work, we hope that given the simplicity of the lower bounding method and its favorable benchmark performance, empirical return based lower bounding can already be a useful contribution, and as a concrete example, it also demonstrates the effectiveness of the lower bounding theory.  Additionally, because future work may propose sampling or other modifications to reduce or remove the bias, we are hesitant to broadly claim that episodic return based lower bounds only work for the deterministic case.  (Appendix A.7 has more about the bias in stochastic environments, which we updated with more details.)

---

### Decision · Program_Chairs · 2023-01-20

**Decision:**

Reject

**Justification For Why Not Higher Score:**

Several concerns on experimentation framework and the significance of the results.

**Justification For Why Not Lower Score:**

N/A

**Metareview: Summary, Strengths And Weaknesses:**

This paper presents a straightforward RL approach that modifies the value target in TD learning. The reviewers appreciate the theoretical analysis and the high quality of writing. However, a significant weakness in this work is the use of SAC for Atari environments. Atari environments require DQN/Rainbow style Q learning approaches for state-of-the-art results, hence several reviewers have raised this point. Perhaps using Mujoco environments, or switching to Rainbow base for Atari will address this concern. Furthermore, it is a little unclear where the improvements from the proposed algorithm actually come from. A comprehensive ablation analysis will significantly clarify this.

Given these concerns, I would highly recommend revising this work with the additional experiments and resubmitting it to the next conference.